# When are Offline Two-Player Zero-Sum Markov Games Solvable?

**Qiwen Cui**
Paul G. Allen School of Computer Science
Engineering
University of Washington
qwcui@cs.washington.edu

**Simon S. Du**
Paul G. Allen School of Computer Science
Engineering
University of Washington
ssdu@cs.washington.edu

## Abstract

We study what dataset assumption permits solving offline two-player zero-sum Markov games. In stark contrast to the offline single-agent Markov decision process, we show that the single strategy concentration assumption is insufficient for learning the Nash equilibrium (NE) strategy in offline two-player zero-sum Markov games. On the other hand, we propose a new assumption named unilateral concentration and design a pessimism-type algorithm that is provably efficient under this assumption. In addition, we show that the unilateral concentration assumption is necessary for learning an NE strategy. Furthermore, our algorithm can achieve minimax sample complexity without any modification for two widely studied settings: dataset with uniform concentration assumption and turn-based Markov games. Our work serves as an important initial step towards understanding offline multi-agent reinforcement learning.

## 1 Introduction

Promising empirical advances have been achieved in reinforcement learning (RL), including mastering the game of Go [Silver et al., 2016], Poker [Brown et al., 2017], real-time strategy games [Vinyals et al., 2019] and robotic control [Kober et al., 2013]. Notably, many of these successes lie in the domain of multi-agent reinforcement learning (MARL). MARL is about multiple agents interacting in a shared environment, and each of them aims to maximize its own long-term reward. During the learning process, each agent not only needs to identify the environment dynamic but also needs to compete/cooperate with other agents. One important subarea of MARL is offline MARL. In many practical scenarios, we only have access to the offline data or it is too expensive to frequently change the policy [Zhang et al., 2021a]. While there are plenty of empirical works on offline MARL [Pan et al., 2021, Jiang and Lu, 2021], the theoretical understanding is still very limited. In this work, we take an initial step towards understanding when offline MARL is provably solvable.

We consider two-player zero-sum Markov games, where two players simultaneously select actions over multiple time steps in a Markovian environment and the first player aims to maximize the total reward while the second player aims to minimize it. In the offline setting, we have access to a fixed dataset collected by a (possibly unknown) exploration policy and the target is to find a (near-)Nash equilibrium (NE) strategy of the underlying two-player zero-sum Markov game.

One of the main difficulties in offline RL is distribution shift, i.e., the dataset distribution is different from the distribution induced by the optimal policy. It is important to understand what is the minimal dataset distribution assumption that permits offline RL. For single-agent offline RL, it is shown that the pessimism principle allows policy optimization with *single policy concentration*, i.e. the dataset only covers the optimal policy [Jin et al., 2021b, Zanette et al., 2021, Yin and Wang, 2021, Rashidinejad et al., 2021]. This assumption is necessary as it is impossible to learn the optimal

policy if it is not covered by the dataset. However, the dataset coverage assumption for MARL is still far from clear. In this work, we want to answer the following question:

*What is the minimal dataset coverage assumption that permits learning an NE strategy in offline two-player zero-sum Markov games?*

Generally speaking, MARL is much more difficult than single-agent RL due to the following two reasons. First, MARL is known to suffer from the *non-stationary* property, i.e. agents will affect the others during the learning process [Zhang et al., 2021a]. Specifically, the performance may decline if each agent simultaneously tries to improve its own policy depending on others' current policies. In addition, multiple agents incur complicated statistical dependence that makes the theoretical analysis difficult. A line of works study Markov games with online sampling oracle [Bai et al., 2020, Bai and Jin, 2020, Liu et al., 2021] or generative model oracle [Sidford et al., 2020, Zhang et al., 2020, Cui and Yang, 2020], where specialized techniques are developed to tackle the above difficulties. In this paper, we give the first analysis on offline Markov games in the fundamental tabular setting.

## 1.1  Main Contributions

● First, we propose an assumption named *unilateral concentration*, which posits that for all strategies $\mu$, $\nu$, strategy pairs $(\mu^*, \nu)$ and $(\mu, \nu^*)$ are covered by the dataset, where $\mu$ is the strategy for the first (max) player, $\nu$ is the strategy for the second (min) player, and $(\mu^*, \nu^*)$ is an NE strategy. In Section 3, we prove that NE strategy is not learnable even if this assumption is only slightly violated. The intuition behind the hardness result is that to identify an NE strategy, the algorithm has to compare it with strategy pairs that one player uses any other strategies as a reference. This result also implies that the single strategy concentration, which is sufficient for offline single-agent RL, is *not sufficient* for offline MARL.

● Second, we provide positive results showing that NE strategy is PAC learnable under the unilateral concentration assumption. Combined with the hardness results above, we conclude that unilateral concentration assumption is the *necessary and sufficient dataset coverage assumption for solving offline zero-sum* Markov games. Our algorithm is based on the pessimism principle that we maintain pessimistic estimates for both players, respectively. We show that our algorithm achieves $\widetilde{O}(\sqrt{C^* SABH^3/n})$ performance gap under unilateral concentration assumption, where $C^*$ quantifies the coverage of the dataset, $S$ is the number of states, $A$ is the number of the max player's actions, $B$ is the number of the min player's actions, $H$ is the horizon and $n$ is the number of samples.

● Third, we show that our algorithm is *minimax optimal* when the dataset satisfies a stronger assumption, uniform concentration, or the Markov game is turn-based. These are two widely studied settings in the RL community. Uniform concentration assumes that all state-action pairs are covered by the dataset and turn-based Markov game is a variant of zero-sum Markov games where two players select actions in turns instead of simultaneously. Although uniform concentration is about the dataset structure and turn-based Markov games are about the environment structure, our algorithm can adapt to both of them without any modification and achieves minimax sample complexity.

**Main Techniques.** Our algorithm is motivated by the Bernstein-type bonus and reference advantage function techniques in Xie et al. [2021b] while we make novel adaptations, namely monotonic update and a self-bounding technique, to realize them in Markov games. The Monotonic update allows a sandwich-type argument that bounds the reference function and further bounds the variance term. The self-bounding technique is utilized to bound the performance gap by itself and then solve the inequality to derive the final bound on performance gap.

To summarize, (1) we identify the minimal dataset coverage assumption that allows learning the NE strategy in Markov games; (2) we propose a pessimism-based algorithm that achieves polynomial sample complexity based on novel Markov game techniques; and (3) we further show the algorithm is minimax optimal under the uniform concentration assumption or in turn-based Markov games.

## 1.2  Related Work

Here we focus on the theoretical works on two-player zero-sum Markov games and offline RL.

**Two-player zero-sum Markov games.** Zero-sum Markov games have been widely studied since the seminal work [Shapley, 1953]. When the transition kernel is unknown, different sampling oracles are utilized to acquire samples, including online sampling [Bai and Jin, 2020, Xie et al., 2020a, Liu et al., 2021, Bai et al., 2020, Jin et al., 2021a, Song et al., 2021], generative model sampling [Sidford et al., 2020, Cui and Yang, 2020, Zhang et al., 2020, Jia et al., 2019]. For offline sampling oracle, Zhang et al. [2021b] and Abe and Kaneko [2020] consider decentralized algorithm with network communication and offline policy evaluation, both under the uniform concentration assumption. One concurrent work [Zhong et al., 2022] considers zero-sum Markov games with linear function approximation. They also show the single policy coverage is not sufficient and propose a similar unilateral concentration assumption under which they give a provably efficient algorithm. On the other hand, under the unilateral concentration assumption, their sample complexity is worse than ours when specialized to tabular setting because they did not use Bernstein bonus. They show it is impossible to learn in all instances without unilateral concentration. However, they do not show that any assumption weaker than unilateral concentration makes learning impossible, which is a negative result proven in our paper. Lastly, our algorithm is minimax optimal for uniform concentration setting and turn-based Markov games while their algorithms are not.

**Offline single-agent RL.** Theoretical analysis of offline RL can be traced back to Szepesvári and Munos [2005], under the uniform concentration assumption (analogue to Assumption 2.3). This assumption has been extensively investigated [Xie and Jiang, 2021, Xie et al., 2020b, Yin et al., 2020, 2021, Ren et al., 2021]. Recently, a line of works showed that the pessimism principle allows offline policy optimization under a much weaker assumption, single policy concentration, both in tabular case and with function approximation [Rashidinejad et al., 2021, Yin and Wang, 2021, Xie et al., 2021b, Jin et al., 2021b, Uehara and Sun, 2021, Uehara et al., 2021, Zanette et al., 2021, Xie et al., 2021a]. One closely related work is Xie et al. [2021b], which utilizes the reference advantage function technique and Bernstein-type bonus to show a minimax sample complexity $\widetilde{O}(SC^*H^3/n)$ in finite-horizon MDP. We show that the counterpart of single policy concentration in zero-sum Markov games is insufficient for NE strategy learning and use the pessimism principle to design algorithm that works under the unilateral concentration assumption.

## 2 Preliminaries

### 2.1 Two-Player Zero-sum Markov Games

Zero-sum Markov games (MG) generalize single-agent MDP to two-agent case where one agent aims to maximize the total reward while the other one aims to minimize it. A tabular finite-horizon zero-sum Markov game is described by the tuple $\mathcal{G} = (\mathcal{S}, \mathcal{A}, \mathcal{B}, P, r, H)$, where $\mathcal{S}$ is the state space, $\mathcal{A}$ is the action space of the first (max) player, $\mathcal{B}$ is the action space of the second (min) player, $P = (P_1, P_2, \cdots, P_H), P_h \in \mathbb{R}^{|\mathcal{S}||\mathcal{A}||\mathcal{B}| \times |\mathcal{S}|}, \forall h \in [H]$ is the (unknown) transition probability matrix for time step $h$, $r = (r_1, r_2, \cdots, r_H), r_h \in [0, 1]^{|\mathcal{S}||\mathcal{A}||\mathcal{B}|}, \forall h \in [H]$ is the (unknown) deterministic reward vector and $H$ is the horizon length.[*] This paper focuses on the tabular setting where $|\mathcal{S}|, |\mathcal{A}|$, and $|\mathcal{B}|$ are finite. At each timestep $h$ and state $s_h$, if the max player chooses action $a_h$ and the min player chooses action $b_h$, then the next state at timestep $h+1$ follows the distribution $s_{h+1} \sim P_h(\cdot|s_h, a_h, b_h)$ and both players receive a reward $r_h(s_h, a_h, b_h)$. Both players sequentially choose $H$ actions and at each timestep, the action is chosen *simultaneously* and then it is revealed to both players. We assume that we have a fixed initial state $s_1$ and it is straightforward to generalize our results to the case where the initial state is sampled from a fixed distribution.[†]

Turn-based Markov games are an important subclass of (simultaneous-move) Markov games, where the max player takes action first and the min player can take action after observing the opponent's action. It is a widely studied setting [Sidford et al., 2020, Cui and Yang, 2020, Bai and Jin, 2020] and we will provide minimax sample complexity result for this setting in Section 4.3.

We denote a strategy pair as $\pi = (\mu, \nu)$, where $\mu = (\mu_1, \mu_2, \cdots, \mu_H), \mu_h : \mathcal{S} \to \Delta^{\mathcal{A}}, \forall h \in [H]$ is the strategy of the first player and $\nu = (\nu_1, \nu_2, \cdots, \nu_H), \nu_h : \mathcal{S} \to \Delta^{\mathcal{B}}, \forall h \in [H]$ is the strategy of

---

[*]It is straightforward to generalize our results to stochastic rewards because the major difficulty is in learning the transitions rather than learning the rewards.

[†]Stochastic initial state is equivalent to an MDP with deterministic initial state by creating a dummy initial state which transits to the next state following that initial state distribution.

the second player, where $\Delta^{\mathcal{X}}$ is the probability simplex on the finite set $\mathcal{X}$. A deterministic strategy is a strategy that maps state to a single point distribution. We define the state value function and state-action value function for a strategy pair $\pi$ similarly as in single-agent MDP:

$$V_h^\pi(s_h) := \mathbb{E}\left[\sum_{t=h}^H r(s_t, a_t, b_t)|\pi, s_h\right], Q_h^\pi(s_h, a_h, b_h) := \mathbb{E}\left[\sum_{t=h}^H r(s_t, a_t, b_t)|\pi, s_h, a_h, b_h\right].$$

If the second player's strategy $\nu$ is fixed, then the MG degenerates to an MDP and we call the optimal policy in this MDP as the best response strategy $\mathrm{br}_1(\nu)$. Similarly, we can define the $\mathrm{br}_2(\mu)$ as the best response for the second player. We will ignore the subscript in $\mathrm{br}_1$ and $\mathrm{br}_2$ when it is clear in the context. While the best response may not be unique, the best response value is always unique. For all $h \in [H], s_h \in \mathcal{S}$, we define

$$V_h^{*,\nu}(s_h) := V_h^{\mathrm{br}(\nu),\nu}(s_h) = \max_\mu V_h^{\mu,\nu}(s_h), V_h^{\mu,*}(s_h) := V_h^{\mu,\mathrm{br}(\mu)}(s_h) = \min_\nu V_h^{\mu,\nu}(s_h).$$

It is well known that Nash equilibrium (NE) strategy $\pi^* = (\mu^*, \nu^*)$, i.e., a strategy pair such that no player can benefit from switching its own strategy, exists for zero-sum Markov games with a unique value function [Shapley, 1953]. In other words, $\mu^*$ and $\nu^*$ are the best responses to each other. We define $V_h^* := V_h^{\mu^*,\nu^*}$ for all $h \in [H]$. The following weak duality property holds for all strategy pairs $(\mu, \nu)$ in MG:

$$V_h^{\mu,*} \le V_h^* \le V_h^{*,\nu}, \forall h \in [H].$$

For a strategy pair $\pi = (\mu, \nu)$, we can then define the corresponding duality gap as

$$\mathrm{Gap}(\pi) = V_1^{*,\nu}(s_1) - V_1^{\mu,*}(s_1).$$

The duality gap is always non-negative and the NE strategy has zero duality gap $\mathrm{Gap}(\pi^*) = 0$. Duality gap measures how well a strategy pair approximates the NE. We say a strategy pair $\pi$ is an $\epsilon$-approximate NE if $\mathrm{Gap}(\pi) \le \epsilon$.

## 2.2 Offline Two-Player Zero-Sum Game

In offline RL, we are given an offline dataset $D = \{(s_h^\tau, a_h^\tau, b_h^\tau, r_h^\tau, s_{h+1}^\tau)\}_{\tau \in [n]}^{h \in [H]}$ and we cannot do any further sampling [Kakade, 2003]. We assume that the dataset is sampled from some exploration policy $\rho = (\rho_1, \rho_2, \cdots, \rho_H), \rho_h : \mathcal{S} \to \Delta^{\mathcal{A} \times \mathcal{B}}, \forall h \in [H]$.[‡] The target of offline MG is to find an approximate NE with a small duality gap by utilizing the given dataset $D$. We use $d_h^\pi(s, a, b)$ to denote the probability of $s, a, b$ appears at timestep $h$ in the trajectory generated by strategy $\pi$ for all $h \in [H]$. The dataset distribution $d_h^\rho(s, a, b)$ is defined similarly. A state-action pair $(s, a, b)$ at timestep $h$ is covered by strategy $\pi$ if and only if $d_h^\pi(s, a, b) > 0$. Strategy $\pi$ is covered by dataset generated by exploration strategy $\rho$ if and only if for all $(s, a, b)$ covered by $\pi$, it is covered by $\rho$. In other words, we have

$$\frac{d_h^\pi(s, a, b)}{d_h^\rho(s, a, b)} < \infty, \forall h \in [H], (s, a, b) \in \mathcal{S} \times \mathcal{A} \times \mathcal{B}. \tag{1}$$

The sample complexity guarantee will depend on this ratio.

**Dataset Coverage Assumptions.** Below we list three different dataset coverage assumptions for Markov games.

**Assumption 2.1.** (Single strategy concentration) One NE strategy $(\mu^*, \nu^*)$ is covered by the dataset.

**Assumption 2.2.** (Unilateral concentration) For all strategies $\mu$ and $\nu$, $(\mu, \nu^*)$ and $(\mu^*, \nu)$ are covered by the dataset, where $(\mu^*, \nu^*)$ is one NE strategy.

**Assumption 2.3.** (Uniform concentration) For all $h \in [H]$ and $(s, a, b) \in \mathcal{S} \times \mathcal{A} \times \mathcal{B}$, $(s, a, b)$ at timestep $h$ is covered by the dataset.

Assumption 2.1 is the weakest assumption and is the most straightforward extension of the single policy concentration in single-agent RL [Rashidinejad et al., 2021]. Assumption 2.3 generalizes the uniform policy concentration in single-agent RL [Yin et al., 2020]. Assumption 2.2 is sandwiched

---

[‡]For simplicity we assume the exploration policy is Markovian. However, our analysis can be directly generalized to arbitrary dataset distribution. See Jin et al. [2021b] for discussions on dataset-dependent bounds.

by Assumption 2.1 and Assumption 2.3 as Assumption 2.2 implies Assumption 2.1 and Assumption 2.3 implies Assumption 2.2. In this work, we will show that Assumption 2.2 is the minimal dataset coverage assumption that allows NE learning and we provide sample complexity bounds that depends on the density ratio (1).[§]

**Notations.** We use $\text{Var}_{P(s,a,b)}(V)$ to denote the variance of the random variable $V(s')$ where $s' \sim P(\cdot|s,a,b)$ and $\text{Var}_P(V) \in \mathbb{R}^{SAB}$ to denote a vector whose $(s,a,b)$ component is $\text{Var}_{P(s,a,b)}(V)$. We define $a \vee b := \max\{a,b\}$ and $a \wedge b := \min\{a,b\}$. In addition, if $a$ is a vector and $b$ is a scalar, the operation is taken on each element of $a$: $[a \vee b]_i = a_i \vee b$. For two vector $a \in \mathbb{R}^n$, $b \in \mathbb{R}^n$, we use $\frac{a}{b} \in \mathbb{R}^n$ to denote the element-wise division: $\left[\frac{a}{b}\right]_i = \frac{a_i}{b_i}$. In addition, if $a$ is scalar, we still use $\frac{a}{b} \in \mathbb{R}^n$ to denote the element-wise division: $\left[\frac{a}{b}\right]_i = \frac{a}{b_i}$. We use $S, A, B$ to denote $|\mathcal{S}|, |\mathcal{A}|, |\mathcal{B}|$.

## 3 Impossibility Results

In this section, we show that no assumption weaker than the unilateral concentration assumption (Assumption 2.2), which includes single strategy concentration (Assumption 2.1), allows learning the NE strategy. To begin with, we consider the deterministic unilateral concentration assumption.

**Assumption 3.1.** (Deterministic unilateral concentration) For all deterministic strategy $\mu$ and $\nu$, $(\mu, \nu^*)$ and $(\mu^*, \nu)$ are covered by the dataset, where $(\mu^*, \nu^*)$ is one NE strategy.

Immediately we can tell that Assumption 3.1 is satisfied under Assumption 2.2. These two assumptions are equivalent, which is shown by Proposition 3.2, because any stochastic strategy can be viewed as a combination of several deterministic strategies.

**Proposition 3.2.** *If for all deterministic strategy $\mu$ and $\nu$, $(\mu, \nu^*)$ and $(\mu^*, \nu)$ are covered by the dataset, then we have for all (possibly stochastic) strategy $\mu'$ and $\nu'$, $(\mu', \nu^*)$ and $(\mu^*, \nu')$ are covered by the dataset.*

For the hardness examples, we consider bandit games, i.e., Markov games with horizon $H = 1$. The result can be generalized to arbitrary horizon by setting the reward to be $0$ in horizons other than $h = 1$. We consider a class of bandit games and datasets such that Assumption 3.1 is almost satisfied while no algorithm can identify the NE strategy for all bandit games and datasets in this class. As Assumption 2.2 and Assumption 3.1 are equivalent, no assumption weaker than Assumption 2.2 allows NE strategy learning. A direct corollary is that single strategy concentration (Assumption 2.1) is not sufficient for NE learning.

**Theorem 3.3.** *Define a class $\mathcal{X}$ of bandit game $M$ and exploration strategy $\rho$ that consists of all $M$ and $\rho$ pairs satisfying that there exists at most one deterministic strategy $\mu$ or one deterministic strategy $\nu$ such that $(\mu, \nu^*)$ or $(\mu^*, \nu)$ is not covered and for all other deterministic strategies $\mu', \nu'$, the density ratio is bounded*

$$\frac{d_h^{\mu^*,\nu'}(s,a,b)}{d_h^\rho(s,a,b)} \le 2A + 2B, \frac{d_h^{\mu',\nu^*}(s,a,b)}{d_h^\rho(s,a,b)} \le 2A + 2B,$$

*for all $h \in [H]$. For any algorithm **ALG**, there exists $(M, \rho) \in \mathcal{X}$ such that the output of the algorithm **ALG** is at most a $0.25$-approximate NE strategy no matter how many data are collected.*

*Proof.* We consider bandit games with two actions for each player here. The action set is $\mathcal{A} = \{a_1, a_2\}$ for the first (max) player and $\mathcal{B} = \{b_1, b_2\}$ for the second (min) player. We construct the following two bandit games with deterministic rewards.

$$\begin{array}{ll} r(a_1, b_1) = 0.25 & r(a_1, b_2) = 0.5 \\ r(a_2, b_1) = 0 & r(a_2, b_2) = 0.75 \end{array}$$
$$\text{Bandit Game 1}$$

Then the (unique) NE of the first bandit game is $(a_1, b_1)$ and the (unique) NE of the second bandit game is $(a_2, b_2)$. Now we set the exploration strategy $\rho$ to be uniform distribution on $\{(a_1, b_1), (a_1, b_2), (a_2, b_2)\}$. We can verify that both bandit games with exploration strategy $\rho$ is

---

[§]Note that there could be different minimal assumptions as the assumption set is a partially ordered set. Here 'minimal' means Assumption 2.2 allows NE learning while no weaker assumption allows doing so.

$$r(a_1, b_1) = 0.25 \quad r(a_1, b_2) = 0.5$$
$$r(a_2, b_1) = 1 \quad\quad r(a_2, b_2) = 0.75$$
$$\text{Bandit Game 2}$$

in the class defined in Theorem 3.3. Note that the dataset contains data on $(a_1, b_1), (a_1, b_2), (a_2, b_2)$ and no data on $(a_2, b_1)$. It is impossible for an algorithm to distinguish between these two bandit games as they are consistent on the given dataset and they all satisfy the dataset coverage assumption that only one action pair is not covered. With some calculations, we can show that the output of **ALG** is at most a 0.25-approximate NE for one of the instances, which proves the theorem. $\square$

*Remark* 3.4. We can easily extend this instance to arbitrary action space by setting $(a_i, b_j) = 0$ for all $i \notin \{1, 2\}, j \in \{1, 2\}$, and $(a_i, b_j) = 1$ for all $j \notin \{1, 2\}, i \in \{1, 2\}$, and the exploration strategy $\rho$ to be the uniform distribution on $(a_i, b_j)$ such that $(i, j) \in \{(i, j) : i \in \{1, 2\} \text{ or } j \in \{1, 2\}, (i, j) \neq (2, 1)\}$.

*Remark* 3.5. It is straightforward to verify that the hard instance in Theorem 3.3 also holds for turn-based Markov games. As a result, no assumption weaker than Assumption 2.2 is sufficient for NE learning in turn-based Markov games.

## 4 Provably Efficient Algorithm under Unilateral Concentration

In this section, we show that it is indeed possible to learn the NE with the unilateral concentration assumption. We propose a novel algorithm called Pessimistic Nash Value Iteration (PNVI), which adapts the pessimism principle in single-agent RL to Markov games. Our sample complexity result depends on the following quantity named *unilateral concentrability*:

**Definition 4.1.** (Unilateral concentrability) For Nash equilibrium $\pi^*$, we define

$$C^* := \min_{\pi^* = (\mu^*, \nu^*)} \max_{h, (s, a, b), \mu, \nu} \left\{ \frac{d_h^{\mu^*, \nu}(s, a, b)}{d_h^{\rho}(s, a, b)}, \frac{d_h^{\mu, \nu^*}(s, a, b)}{d_h^{\rho}(s, a, b)} \right\}.$$

By definition, $C^*$ is finite if Assumption 2.2 is satisfied. For the rest of the paper, $\pi^*$ denotes the Nash equilibrium that achieves the minimum here. Note that $C^*$ is not provided to the algorithm.

### 4.1 Hoeffding-type Algorithm with Data Splitting

To illustrate our main algorithm design ideas, we first propose an algorithm with Hoeffding-type bonus and random data splitting. Given a dataset $\mathcal{D} = \left\{ (s_h^k, a_h^k, b_h^k, r_h^k, s_{h+1}^k) \right\}_{k, h=1}^{n, H}$, we denote $n_h(s, a, b) = \sum_{k=1}^{n} \mathbf{1}\left( (s_h^k, a_h^k, b_h^k) = (s, a, b) \right)$ to be the number of times that $(s, a, b)$ is visited at timestep $h$. We set the empirical reward and the empirical transition kernel as

$$\widehat{r}_h(s, a, b) = r_h(s, a, b), \widehat{P}_h(s'|s, a, b) = \frac{\sum_{k=1}^{n} \mathbf{1}\left( (s_h^k, a_h^k, b_h^k, s_{h+1}^k) = (s, a, b, s') \right)}{\sum_{k=1}^{n} \mathbf{1}\left( (s_h^k, a_h^k, b_h^k) = (s, a, b) \right)}, \quad (2)$$

if $n_h(s, a, b) \geq 1$, and $\widehat{r}_h(s, a, b) = 0$, $\widehat{P}_h(s'|s, a, b) = 1/S$ otherwise. In addition, we use $n_h \in \mathbb{R}^{SAB}$ to denote a vector such that $[n_h]_{s,a,b} = n_h(s, a, b)$.

Now we explain Algorithm 1 in detail. First, we split the dataset $\mathcal{D}$ into $H$ small datasets $\{\mathcal{D}_h\}_{h=1}^{H}$ with the same size. Then we use $\mathcal{D}_h$ to estimate the reward and the transition matrix at timestep $h$. The data splitting scheme is to remove the dependence between each timestep. Then the value function is estimated via a value-iteration-type algorithm. At each timestep, we maintain both optimistic and pessimistic estimates by adding/minusing a Hoeffding-type bonus. We use the following Hoeffding-type bonus:

$$\underline{b}_h(s_h, a_h, b_h) = \overline{b}_h(s_h, a_h, b_h) = 4\sqrt{\frac{H^2 \iota}{n_h(s, a, b) \vee 1}}, \quad (3)$$

where $\iota = \log(HSAB/\delta)$. Then we compute the pessimistic estimate $\overline{Q}$ and $\underline{Q}$:

$$\underline{Q}_h = \left( \widehat{r}_h + (\widehat{P}_h \cdot \underline{V}_{h+1}) - \underline{b}_h \right) \vee 0, \overline{Q}_h = \left( \widehat{r}_h + (\widehat{P}_h \cdot \overline{V}_{h+1}) + \overline{b}_h \right) \wedge (H - h + 1). \quad (4)$$

---
**Algorithm 1** Pessimistic Nash Value Iteration (PNVI)
---

**Input:** Offline dataset $\mathcal{D} = \left\{ (s_h^k, a_h^k, b_h^k, r_h^k, s_{h+1}^k) \right\}_{k,h=1}^{n,H}$. Failure Probability $\delta$.

**Initialization:** Set $\underline{V}_{H+1}(\cdot) = \overline{V}_{H+1}(\cdot) = 0$. Randomly split the dataset $\mathcal{D}$ into $\{\mathcal{D}_h\}_{h=1}^{H}$ with $|\mathcal{D}_h| = n/H$. Set $\widehat{r}_h, \widehat{P}_h, \underline{b}_h$ and $\overline{b}_h$ as (2) and (3) using the dataset $\mathcal{D}_h$ for all $h \in [H]$.
**for** time $h = H, H-1, \dots, 1$ **do**
    Set $\underline{Q}_h(\cdot, \cdot, \cdot)$ and $\overline{Q}_h(\cdot, \cdot, \cdot)$ as (4).
    Compute the NE of $\underline{Q}_h(\cdot, \cdot, \cdot)$ as $(\underline{\mu}_h(\cdot), \underline{\nu}_h(\cdot))$.
    Compute $\underline{V}_h(\cdot) = \mathbb{E}_{a \sim \underline{\mu}_h, b \sim \underline{\nu}_h} \underline{Q}_h(\cdot, a, b)$.
    Compute the NE of $\overline{Q}_h(\cdot, \cdot, \cdot)$ as $(\overline{\mu}_h(\cdot), \overline{\nu}_h(\cdot))$.
    Compute $\overline{V}_h(\cdot) = \mathbb{E}_{a \sim \overline{\mu}_h, b \sim \overline{\nu}_h} \overline{Q}_h(\cdot, a, b)$.
**end for**
Output $\underline{\mu} = (\underline{\mu}_1, \underline{\mu}_2, \cdots, \underline{\mu}_H), \overline{\nu} = (\overline{\nu}_1, \overline{\nu}_2, \cdots, \overline{\nu}_H), \{\underline{V}_h\}_{h=1}^{H}, \{\overline{V}_h\}_{h=1}^{H}$.

---

Pessimistic estimate $\underline{Q}_h$ is for the max player, which mimics the pessimism in single-agent RL. $\overline{Q}_h$ using a positive bonus is for the min player, which is also a kind of pessimism as the min player's target is to minimize the reward. We compute the NE strategy of the matrix game $\underline{Q}(s, \cdot, \cdot)$ and $\overline{Q}(s, \cdot, \cdot)$ respectively and use the NE value to be the state value $\underline{V}(s)$ and $\overline{V}(s)$. Note that we only solve a zero-sum matrix game, which is computationally efficient [Chen and Deng, 2006].

*Remark* 4.2. If we compute an $\epsilon_{\text{NE}}/H$-approximate NE of the matrix game $\underline{Q}(s, \cdot, \cdot)$ and $\overline{Q}(s, \cdot, \cdot)$ at each timestep, then the performance gap will only be enlarged by $\widetilde{O}(\epsilon_{\text{NE}})$.

**Theorem 4.3.** *Suppose Assumption 2.2 holds. For any $0 < \delta < 1$ and strategy $\mu, \nu$, with probability $1 - \delta$, the pessimistic values $\underline{V}_h$ and $\overline{V}_h$ of Algorithm 1 satisfy*

$$\mathbb{E}_{\mu^*,\nu}[V_h^*(s_h) - \underline{V}_h(s_h)] \le \widetilde{O}\left(\sqrt{C^* SABH^5/n}\right), \mathbb{E}_{\mu,\nu^*}[\overline{V}_h(s_h) - V_h^*(s_h)] \le \widetilde{O}\left(\sqrt{C^* SABH^5/n}\right),$$

*for all $h \in [H]$, where $s_h$ is sampled from the trajectory following the strategy in the expectation.*

*Proof Sketch.* For simplicity, we only show the guarantee for the strategy $\underline{\mu}$ of the max player. First, we show that under good concentration event, the pessimistic value $\underline{V}_h$ is always smaller than the best response value of $\underline{\mu}$, i.e.

$$\underline{V}_h(s) \le V_h^{\underline{\mu},*}(s), \forall h \in [H], s \in \mathcal{S}.$$

Second, we show that the performance gap of $\underline{\mu}$ is bounded by the expected sum of bonus under the strategy $\mu^*, \underline{\nu}$, i.e.

$$V_h^*(s) - V_h^{\underline{\mu},*}(s) \le V_h^{\mu^*,\underline{\nu}}(s_h) - \underline{V}_h(s_h) \le 2\mathbb{E}_{\mu^*,\underline{\nu}}\left[\sum_{t=h}^{H} \underline{b}_t(s_t, a_t, b_t) | s_h = s\right].$$

Finally, we define a concatenated strategy $\nu' := (\nu_1, \cdots, \nu_{h-1}, \underline{\nu}_h, \cdots, \underline{\nu}_H)$ and then we have

$$\mathbb{E}_{\mu^*,\nu}[V_h^*(s_h) - \underline{V}_h(s_h)] \le 2\mathbb{E}_{\mu^*,\nu'}\sum_{t=h}^{H} \underline{b}_t(s_t, a_t, b_t).$$

As Assumption 2.2 suggests that $(\mu^*, \nu')$ is well covered by the exploration strategy $\rho$, the expected sum of bonus can be bounded. See Appendix B for details. □

Theorem 4.3 provides polynomial bounds on the error of the value estimates in Algorithm 1. It can directly imply the following performance gap bound. In addition, it provides guarantees for the reference function that will be utilized in the next section.

**Corollary 4.4.** *Suppose Assumption 2.2 holds. For any $0 < \delta < 1$, with probability $1 - \delta$, the output policy $\pi = (\underline{\mu}, \overline{\nu})$ of Algorithm 1 satisfies $\text{Gap}(\pi) \le \widetilde{O}\left(\sqrt{C^* SABH^5/n}\right)$.*

Theorem 4.4 shows that the output strategy of Algorithm 1 is an $\widetilde{O}\left(\sqrt{C^* SABH^5/n}\right)$-approximate NE. The parameter $C^*$ measures how the exploration strategy $\rho$ covers the unilateral strategies $(\mu^*, \nu)$ and $(\mu, \nu^*)$ for all $\mu$ and $\nu$.

## 4.2 Bernstein-type Algorithm with Reference Advantage Function Decomposition

In this section, we will derive an improved performance gap bound $\widetilde{O}\left(\sqrt{C^*SABH^3/n}\right)$. The extra $H^2$ is shaved by using Bernstein-type bonus and reference advantage decomposition technique motivated from Xie et al. [2021b]. However, we want to emphasize that zero-sum Markov games are substantially different from MDP and require novel adaptation, which we will describe later.

Due to the space constraint, we put Algorithm 2 in Appendix A. Algorithm 2 is different from Algorithm 1 in two aspects. First, we use the reference advantage decomposition to remove an $H$ factor. The dataset is split into three subset with equal size $\mathcal{D}_{\text{ref}}$, $\mathcal{D}_0$, $\mathcal{D}_1$, and $\mathcal{D}_1$ is further split into $H$ subset with equal size $\{\mathcal{D}_{h,1}\}_{h=1}^H$. We run algorithm 1 on dataset $\mathcal{D}_{\text{ref}}$ and we can obtain pessimistic value estimate $\underline{V}_{\text{ref}}$ and $\overline{V}_{\text{ref}}$ with guarantees by Theorem 4.3. Then we use dataset $\mathcal{D}_0$ to estimate $P_h \underline{V}_{h+1}^{\text{ref}}$ and dataset $\mathcal{D}_{h,1}$ to estimate $P_h(\underline{V}_{h+1} - \underline{V}_{h+1}^{\text{ref}})$. Second, we use a Bernstein-type bonus to remove another $H$ factor. Our updating formulas of $\underline{Q}_h$ and $\overline{Q}_h$ are

$$\underline{Q}_h = \underline{Q}_h^{\text{ref}} \vee [\widehat{r}_{h,0} + (\widehat{P}_{h,0} \cdot \underline{V}_{h+1}^{\text{ref}}) - \underline{b}_{h,0} + (\widehat{P}_{h,1} \cdot (\underline{V}_{h+1} - \underline{V}_{h+1}^{\text{ref}})) - \underline{b}_{h,1}], \tag{5}$$

$$\overline{Q}_h = \overline{Q}_h^{\text{ref}} \wedge [\widehat{r}_{h,0} + (\widehat{P}_{h,0} \cdot \overline{V}_{h+1}^{\text{ref}}) + \overline{b}_{h,0} + (\widehat{P}_{h,1} \cdot (\overline{V}_{h+1} - \overline{V}_{h+1}^{\text{ref}})) + \overline{b}_{h,1}], \tag{6}$$

where we truncate by the reference function to ensure monotonic update so that $\underline{Q}_h$ and $\overline{Q}_h$ are more accurate pessimistic/optimistic estimate compared with the reference function $\underline{Q}_h^{\text{ref}}$ and $\overline{Q}_h^{\text{ref}}$. The bonus functions are defined as

$$\underline{b}_{h,0} = c\left(\sqrt{\frac{\text{Var}_{\widehat{P}_{h,0}}(\underline{V}_{h+1}^{\text{ref}})\iota}{n_{h,0} \vee 1}} + \frac{H\iota}{n_{h,0} \vee 1}\right), \overline{b}_{h,0} = c\left(\sqrt{\frac{\text{Var}_{\widehat{P}_{h,0}}(\overline{V}_{h+1}^{\text{ref}})\iota}{n_{h,0} \vee 1}} + \frac{H\iota}{n_{h,0} \vee 1}\right), \tag{7}$$

$$\underline{b}_{h,1} = c\left(\sqrt{\frac{\text{Var}_{\widehat{P}_{h,1}}(\underline{V}_{h+1} - \underline{V}_{h+1}^{\text{ref}})\iota}{n_{h,1} \vee 1}} + \frac{H\iota}{n_{h,1} \vee 1}\right), \overline{b}_{h,1} = c\left(\sqrt{\frac{\text{Var}_{\widehat{P}_{h,1}}(\overline{V}_{h+1} - \overline{V}_{h+1}^{\text{ref}})\iota}{n_{h,1} \vee 1}} + \frac{H\iota}{n_{h,1} \vee 1}\right), \tag{8}$$

where $c$ is some universal constant and $\text{Var}_{\widehat{P}_{h,0}}(V)$, $\text{Var}_{\widehat{P}_{h,1}}(V)$, $n_{h,0}$, $n_{h,1}$ are all $SAB$-dimension vectors and the operations are element-wise.

**Theorem 4.5.** *Suppose Assumption 2.2 holds. For any $0 < \delta < 1$ and $n \geq C^*SABH^4$, with probability $1 - \delta$, the output policy $\pi = (\underline{\mu}, \overline{\nu})$ of Algorithm 2 satisfies $\text{Gap}(\pi) \leq \widetilde{O}\left(\sqrt{C^*SABH^3/n}\right)$.*

*Remark 4.6.* $n \geq C^*SABH^4$ serves as the burn-in cost, which is standard in the literature. See a more detailed discussion in Li et al. [2021].

*Proof of Sketch.* For simplicity we only show the guarantee for the strategy $\underline{\mu}$ of the max player. First we show that under good concentration event, the pessimistic value $\underline{V}_h$ is always sandwiched by the reference value $\underline{V}_h^{\text{ref}}$ and the best response value of $\underline{\mu}$, i.e.,

$$\underline{V}_h^{\text{ref}}(s) \leq \underline{V}_h(s) \leq V_h^{\underline{\mu},*}(s), \forall h \in [H], s \in \mathcal{S}.$$

Second, we show that the performance gap of $\underline{\mu}$ is bounded by the expected sum of bonus under the strategy $\mu^*, \underline{\nu}$, i.e.,

$$V_1^*(s_1) - V_1^{\underline{\mu},*}(s_1) \leq V_1^{\mu^*,\underline{\nu}}(s_1) - \underline{V}_1(s_1) \leq 2\mathbb{E}_{\mu^*,\underline{\nu}}\sum_{h=1}^H \left[\underline{b}_{h,0}(s_h, a_h, b_h) + \underline{b}_{h,1}(s_h, a_h, b_h)\right].$$

Then we bound the first term by

$$\mathbb{E}_{\mu^*,\underline{\nu}}\sum_{h=1}^H \underline{b}_{h,0}(s_h, a_h, b_h) \leq \widetilde{O}\left(\sqrt{C^*SABH^3/n} + \sqrt{C^*SABH^3/n}\sqrt{V_1^{\mu^*,\underline{\nu}}(s_1) - \underline{V}_1(s_1)}\right),$$

where $\sqrt{V_1^{\mu^*,\underline{\nu}}(s_1) - \underline{V}_1(s_1)}$ is the square root of the term we want to bound. The second term can be bounded similarly. Finally solving the self-bounding inequality for $V_1^{\mu^*,\underline{\nu}}(s_1) - \underline{V}_1(s_1)$ and we have

$$V_1^*(s_1) - V_1^{\mu,*}(s_1) \leq V_1^{\mu^*,\underline{\nu}}(s_1) - \underline{V}_1(s_1) \leq \widetilde{O}\left(\sqrt{C^*SABH^3/n}\right).$$

We utilizes Theorem 4.3 to provide guarantee for the error of the reference function and $\underline{V}_h^{\mathrm{ref}}(s) \leq \underline{V}_h(s) \leq V_h^{\mu,*}(s)$ to bound the variance of the estimation error. See Appendix C for details. □

As MDP are degenerated Markov games with one player having a fixed action, Markov games inherit the lower bounds of MDP. Comparing with the lower bound $\widetilde{\Omega}\left(\sqrt{C^*SH^3/n}\right)$ [Xie et al., 2021b], our bound is already tight in $C^*, S, H$. The extra $AB$ factor is from the Cauchy-Schwarz inequality and the fact that the NE of zero-sum Markov games can be a mixed strategy while deterministic optimal policy always exists for MDP. It is unknown whether the $AB$ factor is removable and we leave it to future work.

## 4.3 Minimax Optimal Sample Complexity Bounds

In this section, we show that Algorithm 2 directly adapts to two popular settings, i.e. Assumption 2.3 (uniform concentration assumption) and turn-based Markov games. In addition, minimax sample complexity can be achieved under both settings. The proof is deferred to Appendix D.

**Theorem 4.7.** *Set $d_m = \min\{d_h^\rho(s,a,b) : h \in [H], (s,a,b) \in \mathcal{S} \times \mathcal{A} \times \mathcal{B}\}$. Suppose Assumption 2.3 holds. For any $0 < \delta < 1$ and $n \geq H^4/d_m$, with probability $1 - \delta$, the output policy $\pi = (\underline{\mu}, \overline{\nu})$ of Algorithm 2 satisfies $\mathrm{Gap}(\pi) \leq \widetilde{O}\left(\sqrt{H^3/(nd_m)}\right)$.*

This bound has no explicit dependence on $AB$ because the Cauchy-Schwarz inequality can be applied on $d_h^{\mu^*,\underline{\nu}}$ instead of $\sqrt{d_h^{\mu^*,\underline{\nu}}}$ (See the proof of Theorem D.1). As the lower bound $\widetilde{\Omega}\left(\sqrt{H^3/(nd_m)}\right)$ for MDP [Yin and Wang, 2021] is the lower bound for Markov games, Algorithm 2 achieves minimax sample complexity under assumption 2.3.

**Theorem 4.8.** *Suppose Assumption 2.2 holds for a turn-based Markov games. For any $0 < \delta < 1$ and $n \geq C^*SH^4$, with probability $1 - \delta$, the output policy $\pi = (\underline{\mu}, \overline{\nu})$ of Algorithm 2 satisfies $\mathrm{Gap}(\pi) \leq \widetilde{O}\left(\sqrt{C^*SH^3/n}\right)$.*

As the lower bound is $\widetilde{\Omega}\left(\sqrt{C^*SH^3/n}\right)$ [Xie et al., 2021b], Algorithm 2 can achieve the minimax sample complexity for turn-based Markov games under assumption 2.2. The difference is due to turn-based Markov games always have pure NE strategies (See the proof of Theorem D.6).

## 5 Conclusion

In this work, we study the minimal dataset coverage assumption for NE learning in two-player zero-sum Markov games. We show that single strategy concentration is not enough for NE learning. Instead, we find a minimal coverage assumption for NE learning and design an algorithm with sample complexity tight in $C^*, \mathcal{S}, H$ under such assumption based on novel techniques. In addition, the algorithm can achieve minimax sample complexity in certain settings. We believe this work can shed new light on offline MARL.

Here we list several open problems for future work. One direction is to find the minimax sample complexity of offline Markov games under the unilateral concentration. Importantly, it is unclear whether $AB$ factor can be reduced [Bai et al., 2020]. Another direction is to design efficient algorithms for offline MARL with a large number of agents without sample complexity scales exponentially with the number of agents.

## Acknowledgements

This work was supported in part by NSF CCF 2212261, NSF IIS 2143493, NSF DMS 2134106, NSF CCF 2019844 and NSF IIS 2110170.

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
