# A  Algorithm

---
**Algorithm 2** Pessimistic Nash Value Iteration with Reference Advantage Decomposition

---
**Input:** Dataset $\mathcal{D} = \left\{ (s_h^k, a_h^k, b_h^k, r_h^k, s_{h+1}^k) \right\}_{k,h=1}^{n,H}$. Failure Probability $\delta$.

**Initialization:** Randomly split the dataset $\mathcal{D}$ into $\mathcal{D}_{\mathrm{ref}}, \mathcal{D}_0, \{\mathcal{D}_{h,1}\}_{h=1}^H$ with $|\mathcal{D}_{\mathrm{ref}}| = n/3$, $|\mathcal{D}_0| = n/3$, $|\mathcal{D}_{h,1}| = n/(3H)$ for all $h \in [H]$.

Set $\underline{V}_{H+1} = \overline{V}_{H+1} = 0$.

Learn the reference value function $\underline{V}_{\mathrm{ref}}, \overline{V}_{\mathrm{ref}} \leftarrow \mathrm{PNVI}(\mathcal{D}_{\mathrm{ref}})$ (Algorithm 1).

Set $\widehat{P}_{h,0}$ and $\widehat{r}_{h,0}$ as (2) using the dataset $\mathcal{D}_0$ for all $h \in [H]$.

Set $\widehat{P}_{h,1}$ and $\widehat{r}_{h,1}$ as (2) using the dataset $\mathcal{D}_{h,1}$ for all $h \in [H]$.

Set $\underline{b}_{h,0}$ and $\overline{b}_{h,0}$ as (7) using the dataset $\mathcal{D}_0$ for all $h \in [H]$.

**for** time $h = H, H-1, \dots, 1$ **do**

    Set $\underline{b}_{h,1}$ and $\overline{b}_{h,1}$ as (8) using the dataset $\mathcal{D}_{h,1}$ for all $h \in [H]$.

    Set $\underline{Q}_h(\cdot, \cdot, \cdot)$ as (5).

    Compute the NE of $\underline{Q}_h(\cdot, \cdot, \cdot)$ as $(\underline{\mu}_h(\cdot), \underline{\nu}_h(\cdot))$.

    Compute $\underline{V}_h(\cdot) = \mathbb{E}_{a \sim \underline{\mu}_h, b \sim \underline{\nu}_h} \underline{Q}_h(\cdot, a, b)$.

    Set $\overline{Q}_h(\cdot, \cdot, \cdot)$ as (6).

    Compute the NE of $\overline{Q}_h(\cdot, \cdot, \cdot)$ as $(\overline{\mu}_h(\cdot), \overline{\nu}_h(\cdot))$.

    Compute $\overline{V}_h(\cdot) = \mathbb{E}_{a \sim \overline{\mu}_h, b \sim \overline{\nu}_h} \overline{Q}_h(\cdot, a, b)$.

**end for**

**Output:** $\underline{\mu} = (\underline{\mu}_1, \underline{\mu}_2, \cdots, \underline{\mu}_H)$, $\overline{\nu} = (\overline{\nu}_1, \overline{\nu}_2, \cdots, \overline{\nu}_H)$.

---

# B  Proofs in Section 4.1

**Lemma B.1.** *(Concentration) With probability $1 - \delta$, we have*

$$\left| r_h(s,a,b) - \widehat{r}_h(s,a,b) + \left\langle P_h(\cdot|s,a,b) - \widehat{P}_h(\cdot|s,a,b), \underline{V}_{h+1}(\cdot) \right\rangle \right| \leq \underline{b}_h(s,a,b),$$

$$\left| r_h(s,a,b) - \widehat{r}_h(s,a,b) + \left\langle P_h(\cdot|s,a,b) - \widehat{P}_h(\cdot|s,a,b), \overline{V}_{h+1}(\cdot) \right\rangle \right| \leq \overline{b}_h(s,a,b),$$

$$\frac{1}{n_h(s,a,b) \vee 1} \leq \frac{8H\iota}{nd_h^\rho(s,a,b)}.$$

*holds for all $h \in [H]$, $s \in \mathcal{S}$, $a \in \mathcal{A}$ and $b \in \mathcal{B}$. We define this as the good event $\mathcal{G}$.*

*Proof.* We provide the proof for the first argument and the proof for the second argument holds similarly. For all $s,a,b,h$, we have

$$|r_h(s,a,b) - \widehat{r}_h(s,a,b)| \leq H\sqrt{\frac{1}{n_h(s,a,b) \vee 1}},$$

as whenever $n_h(s,a,b) \geq 1$, $\widehat{r}_h(s,a,b) = r_h(s,a,b)$. For the concentration on $\left\langle \widehat{P}(\cdot|s,a,b), \underline{V}_{h+1}(\cdot) \right\rangle$, note that $\underline{V}_{h+1}$ only depends on the dataset $\{\mathcal{D}_t\}_{t=h+1}^H$ while $\widehat{P}_h(\cdot|s,a,b)$ only depends on the dataset $\mathcal{D}_h$, which means they are independent and then Hoeffding's inequality can be applied:

$$\left\langle P_h(\cdot|s,a,b) - \widehat{P}_h(\cdot|s,a,b), \underline{V}_{h+1}(\cdot) \right\rangle \leq 2\sqrt{\frac{H^2\iota}{n_h(s,a,b) \vee 1}}.$$

The second argument holds similarly. For the third argument, the proof is from Lemma B.1 in Xie et al. [2021b]. $\qquad\square$

**Lemma B.2.** *(Pessimism) Under the good event $\mathcal{G}$, we have that $\underline{V}_h(s) \leq V_h^{\underline{\mu},*}(s)$ and $\overline{V}_h(s) \geq V_h^{*,\overline{\nu}}(s)$ hold for all $h \in [H]$ and $s \in \mathcal{S}$.*

*Proof.* We prove this lemma by induction. The inequalities trivially hold for $h = H + 1$. If the inequalities hold for timestep $h + 1$, now we consider timestep $h$. By the definition of $\overline{Q}_h(s, a, b)$, we have

$$
\begin{aligned}
\underline{Q}_h(s, a, b) &= \left( \widehat{r}_h(s, a, b) + (\widehat{P}_h \cdot \underline{V}_{h+1})(s, a, b) - \underline{b}_h(s, a, b) \right) \vee 0 \\
&\leq \left( r(s, a, b) + (P \cdot V_{h+1}^{\mu,*})(s, a, b) \right) \vee 0 \\
&= r(s, a, b) + (P \cdot V_{h+1}^{\mu,*})(s, a, b) \\
&= Q_h^{\mu,*}(s, a, b),
\end{aligned}
$$

where the inequality is from Lemma B.1. With the pessimism on the state-action value function, we can prove the pessimism on the state value function.

$$
\begin{aligned}
\underline{V}_h(s) &= \mathbb{E}_{\underline{\mu}_h, \underline{\nu}_h} \underline{Q}_h(s, a, b) \\
&\leq \mathbb{E}_{\underline{\mu}_h, \mathrm{br}(\underline{\mu}_h)} \underline{Q}_h(s, a, b) \\
&\leq \mathbb{E}_{\underline{\mu}_h, \mathrm{br}(\underline{\mu}_h)} Q_h^{\mu,*}(s, a, b) \\
&= V_h^{\mu,*}(s, a, b),
\end{aligned}
$$

where the first inequality is from the definition of NE and the second inequality is from the pessimism of the state-action value function. The arguments for $\overline{V}_h$ hold similarly. Then by mathematical induction we can prove the lemma. $\qquad\square$

**Lemma B.3.** *Under the good event $\mathcal{G}$, for all $h \in [H]$ and $s_h \in \mathcal{S}$, we have*

$$
V_h^*(s_h) - V_h^{\mu,*}(s_h) \leq V_h^{\mu^*, \nu}(s_h) - \underline{V}_h(s_h) \leq 2\mathbb{E}_{\mu^*, \nu}\left[ \sum_{t=h}^H \underline{b}_t(s_t, a_t, b_t) | s_h \right],
$$

$$
V_h^{*, \overline{\nu}}(s_h) - V_h^*(s_h) \leq \overline{V}_h(s_h) - V_h^{\overline{\mu}, \nu^*}(s_h) \leq 2\mathbb{E}_{\overline{\mu}, \nu^*}\left[ \sum_{t=h}^H \overline{b}_t(s_t, a_t, b_t) | s_h \right].
$$

*Proof.* We prove the first argument and the second argument can be proven similarly. By the definition of NE, we have $V_h^* \leq V_h^{\mu^*, \nu}$. Combined with Lemma B.2, we have the first inequality. For the second inequality, we have

$$
\begin{aligned}
& V_h^{\mu^*, \nu}(s_h) - \underline{V}_h(s_h) \\
={}& \mathbb{E}_{\mu_h^*, \nu_h} Q_h^{\mu^*, \nu}(s_h, a_h, b_h) - \mathbb{E}_{\underline{\mu}_h, \underline{\nu}_h} \underline{Q}_h(s_h, a_h, b_h) \\
\leq{}& \mathbb{E}_{\mu_h^*, \nu_h} Q_h^{\mu^*, \nu}(s_h, a_h, b_h) - \mathbb{E}_{\mu_h^*, \nu_h} \underline{Q}_h(s_h, a_h, b_h) \\
={}& \mathbb{E}_{\mu_h^*, \nu_h} \left[ Q_h^{\mu^*, \nu}(s_h, a_h, b_h) - \underline{Q}(s_h, a_h, b_h) \right] \\
={}& \mathbb{E}_{\mu_h^*, \nu_h} \left[ r_h(s_h, a_h, b_h) + \left\langle P_h(\cdot|s_h, a_h, b_h), V_{h+1}^{\mu^*, \nu}(\cdot) \right\rangle - \widehat{r}_h(s_h, a_h, b_h) \right. \\
& \left. - \left\langle \widehat{P}_h(\cdot|s_h, a_h, b_h), \underline{V}_{h+1}(\cdot) \right\rangle + \underline{b}_h(s_h, a_h, b_h) \right] \\
\leq{}& \mathbb{E}_{\mu_h^*, \nu_h} \left[ \left\langle P_h(\cdot|s_h, a_h, b_h), V_{h+1}^{\mu^*, \nu}(\cdot) - \underline{V}_{h+1}(\cdot) \right\rangle + 2\underline{b}_h(s_h, a_h, b_h) \right] \qquad \text{(Lemma B.1)} \\
={}& \mathbb{E}_{\mu_h^*, \nu_h} \left[ V_{h+1}^*(s_{h+1}) - \underline{V}_{h+1}^*(s_{h+1}) | s_h \right] + 2\mathbb{E}_{\mu_h^*, \nu_h^*} \underline{b}_h(s_h, a_h, b_h) \\
\leq{}& 2\mathbb{E}_{\mu^*, \nu} \left[ \sum_{t=h}^H \underline{b}_h(s_t, a_t, b_t) | s_h \right].
\end{aligned}
$$

$\qquad\square$

**Theorem B.4.** *Suppose Assumption 2.2 holds. For any $0 < \delta < 1$, with probability $1 - \delta$, the output policy $\pi = (\underline{\mu}, \overline{\nu})$ of Algorithm 1 satisfies*

$$V_1^*(s_1) - V_1^{\underline{\mu},*}(s_1) \leq 64\sqrt{\frac{C^* SABH^5\iota^2}{n}}, V_1^{*,\overline{\nu}}(s_1) - V_1^*(s_1) \leq 64\sqrt{\frac{C^* SABH^5\iota^2}{n}}.$$

*As a result, we have*

$$\mathrm{Gap}(\underline{\mu}, \overline{\nu}) \leq \widetilde{O}\left(\sqrt{\frac{C^* SABH^5}{n}}\right).$$

*Proof.* By Lemma B.3, with probability at least $1 - \delta$, we have

$$V_1^{\mu^*,*}(s_1) - V_1^{\underline{\mu},*}(s_1)$$

$$\leq 2\sum_{h=1}^H \mathbb{E}_{\mu^*,\underline{\nu}}\underline{b}_h(s_h, a_h, b_h)$$

$$= 2\sum_{h=1}^H \mathbb{E}_{\mu^*,\underline{\nu}}\left[4\sqrt{\frac{H^2\iota}{n_h(s,a,b) \vee 1}}\right]$$

$$\leq 2\sum_{h=1}^H \mathbb{E}_{\mu^*,\underline{\nu}}\left[32\sqrt{\frac{H^3\iota^2}{nd_h^\rho(s,a,b)}}\right] \qquad \text{(Lemma B.1)}$$

$$= 2\sum_{h=1}^H \sum_{(s,a,b)} d_h^{\mu^*,\underline{\nu}}(s,a,b)\left[32\sqrt{\frac{H^3\iota^2}{nd_h^\rho(s,a,b)}}\right]$$

$$\leq 64\sum_{h=1}^H \sum_{(s,a,b)}\left[\sqrt{\frac{d_h^{\mu^*,\underline{\nu}}(s,a,b)C^* H^3\iota^2}{n}}\right]$$

$$\leq 64\sqrt{SABH} \cdot \sqrt{\frac{\sum_{h=1}^H \sum_{(s,a,b)} d_h^{\mu^*,\underline{\nu}}(s,a,b)C^* H^3\iota^2}{n}} \qquad \text{(Cauchy-Schwarz Inequality)}$$

$$= 64\sqrt{\frac{C^* SABH^5\iota^2}{n}}.$$

Similarly we have

$$V_1^{*,\overline{\nu}}(s_1) - V_1^*(s_1) \leq 64\sqrt{\frac{C^* SABH^5\iota^2}{n}}.$$

As a result, we have

$$\mathrm{Gap}(\underline{\mu}, \overline{\nu}) \leq V_1^{*,\overline{\nu}}(s_1) - V_1^*(s_1) + V_1^{\mu^*,*}(s_1) - V_1^{\underline{\mu},*}(s_1) \leq \widetilde{O}\left(\sqrt{\frac{C^* SABH^5}{n}}\right).$$

$\square$

**Theorem B.5.** *Suppose Assumption 2.2 holds. For any $0 < \delta < 1$ and strategy $\mu, \nu$, with probability $1 - \delta$, the pessimistic values $\underline{V}_h$ and $\overline{V}_h$ of Algorithm 1 satisfy*

$$\mathbb{E}_{\mu^*,\nu}\left[V_h^*(s_h) - \underline{V}_h(s_h)\right] \leq 64\sqrt{\frac{C^* SABH^5\iota^2}{n}},$$

$$\mathbb{E}_{\mu,\nu^*}\left[\overline{V}_h(s_h) - V_h^*(s_h)\right] \leq 64\sqrt{\frac{C^* SABH^5\iota^2}{n}},$$

*where $s_h$ is sampled from the trajectory following the strategy in the expectation at timestep $h$.*

*Proof.* We prove the first argument and the second argument can be proven similarly. By Lemma B.3, under good event $\mathcal{G}$ for all state $s$ we have

$$V_h^*(s) - V_h^{\underline{\mu},*}(s)$$

$$\leq 2 \sum_{t=h}^{H} \mathbb{E}_{\mu^*, \underline{\nu}} \left[ \underline{b}_h(s_t, a_t, b_t) | s_h = s \right].$$

We define $\nu' = (\nu_1, \cdots, \nu_{h-1}, \underline{\nu}_h, \cdots, \underline{\nu}_H)$. Then we have

$$\mathbb{E}_{\mu^*, \nu} \left[ V_h^*(s_h) - \underline{V}_h(s_h) \right] \leq \mathbb{E}_{\mu^*, \nu} \left[ 2 \sum_{t=h}^{H} \mathbb{E}_{\mu^*, \underline{\nu}} \left[ \underline{b}_h(s_t, a_t, b_t) | s_h = s \right] | s \right]$$

$$= 2 \sum_{t=h}^{H} \mathbb{E}_{\mu^*, \nu'} \left[ \underline{b}_h(s_t, a_t, b_t) \right].$$

Then following the proof of Theorem B.4, we can prove the argument. □

## C  Proofs in Section 4.2

For simplicity, we only provide the guarantee for the max player and the guarantee for the min player can be proven in a similar manner.

**Lemma C.1.** *(Concentration) There exists some absolute constant $c > 0$ such that the concentration event $\mathcal{G}'$ holds with probability at least $1 - \delta$, i.e.,*

$$\left| \widehat{r}_{h,0}(s, a, b) - r_{h,0}(s, a, b) + \left[ \left( \widehat{P}_{h,0} - P_h \right) V_{h+1}^{\text{ref}} \right] (s, a, b) \right|$$

$$\leq c \left( \sqrt{ \frac{\text{Var}_{\widehat{P}_{h,0}(s,a,b)}(V_{h+1}^{\text{ref}}) \iota}{n_{h,0}(s, a, b) \vee 1} } + \frac{H \iota}{n_{h,0}(s, a, b) \vee 1} \right),$$

$$\left| \left[ \left( \widehat{P}_{h,1} - P_h \right) \left( \underline{V}_{h+1} - \underline{V}_{h+1}^{\text{ref}} \right) \right] (s, a, b) \right|$$

$$\leq c \left( \sqrt{ \frac{\text{Var}_{\widehat{P}_{h,1}(s,a,b)}(\underline{V}_{h+1} - \underline{V}_{h+1}^{\text{ref}}) \iota}{n_{h,1}(s, a, b) \vee 1} } + \frac{H \iota}{n_{h,1}(s, a, b) \vee 1} \right),$$

$$\frac{1}{n_{h,0}(s, a, b) \vee 1} \leq c \frac{\iota}{n d_h^{\rho}(s, a, b)}, \quad \frac{1}{n_{h,1}(s, a, b) \vee 1} \leq c \frac{H \iota}{n d_h^{\rho}(s, a, b)}.$$

*Proof.* The proof is a direct application of Lemma C.1 in Xie et al. [2021b] with $s, a$ replaced by $s, a, b$. □

**Lemma C.2.** *For all $h \in [H]$ and $s \in \mathcal{S}$, we have $\underline{V}_h(s) \geq \underline{V}_h^{\text{ref}}(s)$.*

*Proof.* By the update rule (5), we have $\underline{Q}_h(s, a, b) \geq \underline{Q}_h^{\text{ref}}(s, a, b)$ for $h \in [H]$ and $s, a, b \in \mathcal{S} \times \mathcal{A} \times \mathcal{B}$. Then by the definition of NE, we have

$$\underline{V}_h(s) = \mathbb{E}_{\underline{\mu}_h, \underline{\nu}_h} \underline{Q}_h(s, a, b) \geq \mathbb{E}_{\underline{\mu}_h^{\text{ref}}, \underline{\nu}_h} \underline{Q}_h(s, a, b) \geq \mathbb{E}_{\underline{\mu}_h^{\text{ref}}, \underline{\nu}_h} \underline{Q}_h^{\text{ref}}(s, a, b) \geq \mathbb{E}_{\underline{\mu}_h^{\text{ref}}, \underline{\nu}_h^{\text{ref}}} \underline{Q}_h^{\text{ref}}(s, a, b) = \underline{V}_h^{\text{ref}}(s).$$

□

**Lemma C.3.** *(Pessimism) Under the good event $\mathcal{G}'$, we have that $\underline{V}_h(s) \leq V_h^{\mu,*}(s)$ holds for all $h \in [H]$ and $s \in \mathcal{S}$.*

*Proof.* We prove this lemma by induction. The inequalities trivially hold for $h = H + 1$. If the inequalities hold for $h + 1$, now we consider $h$.

$$\underline{Q}_h(s, a, b)$$
$$= \left\{ \widehat{r}_{h,0}(s, a, b) + (\widehat{P}_{h,0} \cdot \underline{V}_{h+1}^{\text{ref}})(s, a, b) - \underline{b}_{h,0}(s, a, b) + (\widehat{P}_{h,1} \cdot (\underline{V}_{h+1} - \underline{V}_{h+1}^{\text{ref}}))(s, a, b) - \underline{b}_{h,1}(s, a, b) \right\}$$

$$\vee \underline{Q}_h^{\mathrm{ref}}(s,a,b)$$

$$\leq \max\left\{r_h(s,a,b) + (P_h \cdot \underline{V}_{h+1}^{\mathrm{ref}})(s,a,b) + \left(P_h \cdot \left(\underline{V}_{h+1} - \underline{V}_{h+1}^{\mathrm{ref}}\right)\right)(s,a,b), \underline{Q}_h^{\mathrm{ref}}(s,a,b)\right\}$$

$$= \max\left\{r_h(s,a,b) + (P_h \cdot \underline{V}_{h+1})(s,a,b), \underline{Q}_h^{\mathrm{ref}}(s,a,b)\right\}$$

$$\leq \max\left\{r_h(s,a,b) + (P_h \cdot \underline{V}_{h+1})(s,a,b), r_h(s,a,b) + (P_h \cdot \underline{V}_{h+1}^{\mathrm{ref}})(s,a,b)\right\} \qquad \text{(Lemma B.2)}$$

$$\leq r_h(s,a,b) + (P_h \cdot \underline{V}_{h+1})(s,a,b) \qquad\qquad\qquad\qquad\qquad\qquad \text{(Lemma C.2)}$$

$$\leq r_h(s,a,b) + (P_h \cdot V_{h+1}^{\mu,*})(s,a,b) \qquad\qquad\qquad\qquad\qquad \text{(Induction hypothesis)}$$

$$= Q_h^{\mu,*}(s,a,b).$$

Then by the definition of NE, we have

$$\begin{aligned}
\underline{V}_h(s) &= \mathbb{E}_{\underline{\mu}_h, \underline{\nu}_h} \underline{Q}_h(s,a,b) \\
&\leq \mathbb{E}_{\underline{\mu}_h, \mathrm{br}(\underline{\mu}_h)} \underline{Q}_h(s,a,b) \\
&\leq \mathbb{E}_{\underline{\mu}_h, \mathrm{br}(\underline{\mu}_h)} Q_h^{\mu,*}(s,a,b) \\
&= V_h^{\mu,*}(s).
\end{aligned}$$

With mathematical induction we can prove the lemma. $\qquad\square$

**Lemma C.4.** *Under the good event $\mathcal{G}'$, we have*

$$V_1^{\mu^*,\nu}(s_1) - \underline{V}_1(s_1) \leq 2\mathbb{E}_{\mu^*,\nu} \sum_{h=1}^{H} \underline{b}_{h,0}(s_h, a_h, b_h) + 2\mathbb{E}_{\mu^*,\nu} \sum_{h=1}^{H} \underline{b}_{h,1}(s_h, a_h, b_h)$$

*Proof.*

$$V_1^{\mu^*,\nu}(s_1) - \underline{V}_1(s_1)$$

$$= \mathbb{E}_{\mu_1^*,\nu_1} Q_1^{\mu^*,\nu}(s_1, a_1, b_1) - \mathbb{E}_{\underline{\mu}_1, \underline{\nu}_1} \underline{Q}_1(s_1, a_1, b_1)$$

$$\leq \mathbb{E}_{\mu_1^*,\nu_1} Q_1^{\mu^*,\nu}(s_1, a_1, b_1) - \mathbb{E}_{\mu_1^*,\nu_1} \underline{Q}_1(s_1, a_1, b_1)$$

$$= \mathbb{E}_{\mu_1^*,\nu_1} \left[ Q_1^{\mu^*,\nu}(s_1, a_1, b_1) - \underline{Q}_1(s_1, a_1, b_1) \right]$$

$$= \mathbb{E}_{\mu_1^*,\nu_1} \left[ r_1(s_1, a_1, b_1) + \left\langle P_1(\cdot|s_1, a_1, b_1), V_2^{\mu^*,\nu}(\cdot) \right\rangle - \underline{V}_1^{\mathrm{ref}}(s_1) \vee \left\{ \widehat{r}_{1,0}(s_1, a_1, b_1) \right.\right.$$
$$\left.\left. + (\widehat{P}_{1,0} \underline{V}_2^{\mathrm{ref}})(s_1, a_1, b_1) - \underline{b}_{1,0}(s_1, a_1, b_1) + (\widehat{P}_{1,1}(\underline{V}_2 - \underline{V}_2^{\mathrm{ref}}))(s_1, a_1, b_1) - \underline{b}_{1,1}(s_1, a_1, b_1) \right\} \right]$$

$$\leq \mathbb{E}_{\mu_1^*,\nu_1} \left[ \left\langle P_1(\cdot|s_1, a_1, b_1), V_2^{\mu^*,\nu}(\cdot) - \underline{V}_2(\cdot) \right\rangle + 2\underline{b}_{1,0}(s_1, a_1, b_1) + 2\underline{b}_{1,1}(s_1, a_1, b_1) \right]$$
$$\text{(Lemma C.1)}$$

$$= \mathbb{E}_{\mu_1^*,\nu_1} \left[ V_2^{\mu^*,\nu}(s_2) - \underline{V}_2^*(s_2) \right] + 2\mathbb{E}_{\mu_1^*,\nu_1^*} \underline{b}_{1,0}(s_1, a_1, b_1) + 2\mathbb{E}_{\mu_1^*,\nu_1^*} \underline{b}_{1,1}(s_1, a_1, b_1)$$

$$\leq 2\mathbb{E}_{\mu^*,\nu} \sum_{h=1}^{H} \underline{b}_{h,0}(s_h, a_h, b_h) + 2\mathbb{E}_{\mu^*,\nu} \sum_{h=1}^{H} \underline{b}_{h,1}(s_h, a_h, b_h),$$

where the last inequality is from telescoping the timestep $H$. $\qquad\square$

**Lemma C.5.** *For any strategy $\nu$, we have*

$$\sum_{h=1}^{H} \sum_{(s,a,b)} d_h^{\mu^*,\nu}(s,a,b) \operatorname*{Var}_{P_h(s,a,b)}(V_{h+1}^{\mu^*,\nu}) \leq H^2.$$

*Proof.* This is the standard total variance lemma.

$$\sum_{h=1}^{H} \sum_{(s,a,b)} d_h^{\mu^*,\nu}(s,a,b) \operatorname*{Var}_{P_h(s,a,b)}(V_h^{\mu^*,\nu})$$

$$= \sum_{h=1}^{H} \mathbb{E}_{\mu^*,\nu} \left[ \text{Var} \left[ V_{h+1}^*(s_{h+1}) | s_h, a_h, b_h \right] \right]$$

$$= \sum_{h=1}^{H} \mathbb{E}_{\mu^*,\nu} \left[ \mathbb{E} \left[ \left( V_{h+1}^*(s_{h+1}) + r_h(s_h, a_h, b_h) - V_h^*(s_h) \right)^2 | s_h, a_h, b_h \right] \right]$$

$$= \sum_{h=1}^{H} \mathbb{E}_{\mu^*,\nu} \left[ \left( V_{h+1}^*(s_{h+1}) + r_h(s_h, a_h, b_h) - V_h^*(s_h) \right)^2 \right]$$

$$= \mathbb{E}_{\mu^*,\nu} \left[ \left( \sum_{h=1}^{H} \left( V_{h+1}^*(s_{h+1}) + r_h(s_h, a_h, b_h) - V_h^*(s_h) \right) \right)^2 \right]$$

$$= \mathbb{E}_{\mu^*,\nu} \left[ \left( \sum_{h=1}^{H} r_h(s_h, a_h, b_h) - V_1^*(s_1) \right)^2 \right]$$

$$= \text{Var}_{\mu^*,\nu} \left( \sum_{h=1}^{H} r_h(s_h, a_h, b_h) \right)$$

$$\leq H^2.$$

$\qquad\qquad\qquad\qquad\qquad\qquad\qquad\qquad\qquad\qquad\qquad\qquad\qquad\qquad\qquad\qquad\qquad\qquad$ □

**Lemma C.6.** *The output strategy $\pi = (\underline{\mu}, \overline{\nu})$ and the pessimistic estimate $\underline{V}$ of Algorithm 1 satisfy*

$$V_1^{\mu^*,\underline{\nu}}(s_1) - \underline{V}_1(s_1) \geq \mathbb{E}_{\mu^*,\underline{\nu}} \left[ V_h^{\mu^*,\underline{\nu}}(s_h) - \underline{V}_h(s_h) \right].$$

*Proof.* We prove the argument for $h = 2$ first.

$$V_1^{\mu^*,\underline{\nu}}(s_1) - \underline{V}_1(s_1)$$

$$\geq \mathbb{E}_{\mu^*,\underline{\nu}} [Q_1^{\mu^*,\underline{\nu}}(s_1, a_1, b_1) - \underline{Q}_1(s_1, a_1, b_1)]$$

$$\geq \mathbb{E}_{\mu^*,\underline{\nu}} \left[ r_1(s_1, a_1, b_1) + \left\langle P_1(\cdot|s_1, a_1, b_1), V_2^{\mu^*,\underline{\nu}}(\cdot) \right\rangle \right] - \mathbb{E}_{\mu^*,\underline{\nu}} \left[ \widehat{r}_{1,0}(s_1, a_1, b_1) + (\widehat{P}_{1,0} \underline{V}_2^{\text{ref}})(s_1, a_1, b_1) \right.$$

$$\left. - \underline{b}_{1,0}(s_1, a_1, b_1) + (\widehat{P}_{1,1}(\underline{V}_2 - \underline{V}_2^{\text{ref}}))(s_1, a_1, b_1) - \underline{b}_{1,1}(s_1, a_1, b_1) \right]$$

$$\geq \mathbb{E}_{\mu^*,\underline{\nu}} \left[ r_1(s_1, a_1, b_1) + \left\langle P_1(\cdot|s_1, a_1, b_1), V_2^{\mu^*,\underline{\nu}}(\cdot) \right\rangle \right] - \mathbb{E}_{\mu^*,\underline{\nu}} \left[ r_1(s_1, a_1, b_1) + \left\langle P_1(\cdot|s_1, a_1, b_1), \underline{V}_2(\cdot) \right\rangle \right]$$

$$= \mathbb{E}_{\mu^*,\underline{\nu}} \left[ V_2^{\mu^*,\underline{\nu}}(s_2) - \underline{V}_2(s_2) \right].$$

We can prove the lemma for arbitrary $h$ by telescoping the argument to timestep $h$.

$\qquad\qquad\qquad\qquad\qquad\qquad\qquad\qquad\qquad\qquad\qquad\qquad\qquad\qquad\qquad\qquad\qquad\qquad$ □

**Lemma C.7.** *For $n \geq C^* SABH^3$, we have*

$$\mathbb{E}_{\mu^*,\underline{\nu}} \sum_{h=1}^{H} \underline{b}_{h,0}(s_h, a_h, b_h) \leq \widetilde{O} \left( \sqrt{\frac{C^* SABH^3}{n}} \sqrt{V_1^{\mu^*,\underline{\nu}}(s_1) - \underline{V}_1(s_1)} \right) + \widetilde{O} \left( \sqrt{\frac{C^* SABH^3}{n}} \right).$$

*Proof.*

$$\mathbb{E}_{\mu^*,\underline{\nu}} \sum_{h=1}^{H} \underline{b}_{h,0}(s_h, a_h, b_h)$$

$$= c \mathbb{E}_{\mu^*,\underline{\nu}} \sum_{h=1}^{H} \left( \sqrt{\frac{\text{Var}_{\widehat{P}_{h,0}(s,a,b)}(\underline{V}_{h+1}^{\text{ref}}) \iota}{n_{h,0}(s, a, b) \vee 1}} + \frac{H\iota}{n_{h,0}(s, a, b) \vee 1} \right)$$

$$\leq c\mathbb{E}_{\mu^*,\underline{\nu}} \sum_{h=1}^{H} \left( \sqrt{\frac{c\,\mathrm{Var}_{P_h(s,a,b)}(\underline{V}_{h+1}^{\mathrm{ref}})\iota}{nd_h^\rho(s,a,b)}} + \frac{cH\iota}{nd_h^\rho(s,a,b)} + \frac{cH\iota}{nd_h^\rho(s,a,b)} \right)$$

$$= c^2 \sum_{h=1}^{H} \sum_{(s,a,b)} d_h^{\mu^*,\underline{\nu}}(s,a,b) \left( \sqrt{\frac{\mathrm{Var}_{P_h(s,a,b)}(\underline{V}_{h+1}^{\mathrm{ref}})\iota}{nd_h^\rho(s,a,b)}} + \frac{H\iota}{nd_h^\rho(s,a,b)} \right)$$

$$\leq c^2 \sum_{h=1}^{H} \sum_{(s,a,b)} \left( \sqrt{\frac{C^* d_h^{\mu^*,\underline{\nu}}(s,a,b)\,\mathrm{Var}_{P_h(s,a,b)}(\underline{V}_{h+1}^{\mathrm{ref}})\iota}{n}} + \frac{C^* H\iota}{n} \right)$$

$$\leq c^2 \sqrt{SABH} \cdot \sqrt{\frac{C^* \iota \sum_{h=1}^{H} \sum_{(s,a,b)} d_h^{\mu^*,\underline{\nu}}(s,a,b)\,\mathrm{Var}_{P_h(s,a,b)}(\underline{V}_{h+1}^{\mathrm{ref}})}{n}} + \frac{c^2 SABC^* H\iota}{n}$$

$$\leq c^2 \sqrt{C^* SABH\iota} \cdot \sqrt{\frac{\sum_{h=1}^{H} \mathbb{E}_{\mu^*,\underline{\nu}} \left[ \mathrm{Var}_{P_h(s,a,b)}(\underline{V}_{h+1}^{\mathrm{ref}}) \right]}{n}} + \frac{c^2 SABC^* H\iota}{n}$$

$$\leq c^2 \sqrt{C^* SABH\iota} \cdot \sqrt{\frac{\sum_{h=1}^{H} \mathbb{E}_{\mu^*,\underline{\nu}} \left[ \mathrm{Var}_{P_h(s,a,b)}(V_{h+1}^{\mu^*,\underline{\nu}}) + 2H[P_h(V_{h+1}^{\mu^*,\underline{\nu}} - \underline{V}_{h+1}^{\mathrm{ref}})](s,a,b) \right]}{n}} + \frac{c^2 SABC^* H\iota}{n}$$

(Lemma E.4)

$$\leq c^2 \sqrt{C^* SABH\iota} \cdot \sqrt{\frac{H^2 + 2H \sum_{h=1}^{H} \mathbb{E}_{\mu^*,\underline{\nu}} \left[ V_{h+1}^{\mu^*,\underline{\nu}}(s_{h+1}) - \underline{V}_{h+1}^{\mathrm{ref}}(s_{h+1}) \right]}{n}} + \frac{c^2 SABC^* H\iota}{n}$$

(Lemma C.5)

$$= c^2 \sqrt{C^* SABH\iota} \cdot \sqrt{\frac{H^2 + 2H \sum_{h=1}^{H} \mathbb{E}_{\mu^*,\underline{\nu}} \left[ V_{h+1}^{\mu^*,\underline{\nu}}(s_{h+1}) - V_{h+1}^{*}(s_{h+1}) + V_{h+1}^{*}(s_{h+1}) - \underline{V}_{h+1}^{\mathrm{ref}}(s_{h+1}) \right]}{n}}$$
$$+ \frac{c^2 SABC^* H\iota}{n}$$

$$\leq c^2 \sqrt{C^* SABH\iota} \cdot \sqrt{\frac{H^2 + 2H^2(V_1^{\mu^*,\underline{\nu}}(s_1) - \underline{V}_1(s_1)) + 128H\sqrt{\frac{C^* SABH^5\iota^2}{n_{\mathrm{ref}}}}}{n}} + \frac{c^2 SABC^* H\iota}{n}$$

(Lemma C.6 and Theorem B.5)

$$\leq \frac{c^2 \sqrt{C^* SABH^3\iota}}{\sqrt{n}} + \frac{c^2 \sqrt{384 C^* SABH^2\iota \sqrt{C^* SABH^5\iota^2}}}{n^{3/4}} + \frac{c\sqrt{2C^* SABH^3\iota}}{\sqrt{n}} \sqrt{V_1^{\mu^*,\underline{\nu}}(s_1) - \underline{V}_1(s_1)}$$
$$+ \frac{c^2 SABC^* H\iota}{n}$$

$$\leq \widetilde{O}\left( \sqrt{\frac{C^* SABH^3}{n}} \sqrt{V_1^{\mu^*,\underline{\nu}}(s_1) - \underline{V}_1(s_1)} \right) + \widetilde{O}\left( \sqrt{\frac{C^* SABH^3}{n}} \right). \qquad (n \geq C^* SABH^3)$$

$\square$

**Lemma C.8.** *For $n \geq C^* SABH^4$, we have*

$$\mathbb{E}_{\mu^*,\underline{\nu}} \sum_{h=1}^{H} \underline{b}_{h,1}(s_h, a_h, b_h) \leq \widetilde{O}\left( \sqrt{\frac{C^* SABH^3}{n}} \right).$$

*Proof.*

$$\mathbb{E}_{\mu^*,\underline{\nu}} \sum_{h=1}^{H} \underline{b}_{h,1}(s_h, a_h, b_h)$$

$$=c\mathbb{E}_{\mu^*,\underline{\nu}}\sum_{h=1}^{H}\left(\sqrt{\frac{\mathrm{Var}_{\widehat{P}_{h,0}(s,a,b)}(\underline{V}_{h+1}-\underline{V}_{h+1}^{\mathrm{ref}})\iota}{n_{h,1}(s,a,b)\vee 1}}+\frac{H\iota}{n_{h,1}(s,a,b)\vee 1}\right)$$

$$\leq c\mathbb{E}_{\mu^*,\underline{\nu}}\sum_{h=1}^{H}\left(\sqrt{\frac{cH\,\mathrm{Var}_{P_h(s,a,b)}(\underline{V}_{h+1}-\underline{V}_{h+1}^{\mathrm{ref}})\iota}{nd_h^{\rho}(s,a,b)}}+\frac{cH^2\iota}{nd_h^{\rho}(s,a,b)}+\frac{cH^2\iota}{nd_h^{\rho}(s,a,b)}\right)$$

$$\leq c^2\mathbb{E}_{\mu^*,\underline{\nu}}\sum_{h=1}^{H}\left(\sqrt{\frac{H\left[P_h(\underline{V}_{h+1}-\underline{V}_{h+1}^{\mathrm{ref}})^2\right](s,a,b)\iota}{nd_h^{\rho}(s,a,b)}}+\frac{H^2\iota}{nd_h^{\rho}(s,a,b)}\right)$$

$$=c^2\sum_{h=1}^{H}\sum_{(s,a,b)}d_h^{\mu^*,\underline{\nu}}(s,a,b)\left(\sqrt{\frac{H\left[P_h(\underline{V}_{h+1}-\underline{V}_{h+1}^{\mathrm{ref}})^2\right](s,a,b)\iota}{nd_h^{\rho}(s,a,b)}}+\frac{H^2\iota}{nd_h^{\rho}(s,a,b)}\right)$$

$$\leq c^2\sum_{h=1}^{H}\sum_{(s,a,b)}\left(\sqrt{\frac{C^*Hd_h^{\mu^*,\underline{\nu}}(s,a,b)\left[P_h(\underline{V}_{h+1}-\underline{V}_{h+1}^{\mathrm{ref}})^2\right](s,a,b)\iota}{n_1}}+\frac{H^2C^*\iota}{n_1}\right)$$

$$\text{(Cauchy-Schwarz Inequality)}$$

$$\leq c^2\sqrt{SABH\iota}\sqrt{\frac{C^*H\sum_{h=1}^{H}\sum_{(s,a,b)}d_h^{\mu^*,\underline{\nu}}(s,a,b)\left[P_h(\underline{V}_{h+1}-\underline{V}_{h+1}^{\mathrm{ref}})^2\right](s,a,b)}{n}}+\frac{c^2SABH^3C^*\iota}{n}$$

$$\leq c^2\sqrt{SABH\iota}\sqrt{\frac{C^*H\iota\sum_{h=1}^{H}\sum_{(s,a,b)}d_h^{\mu^*,\underline{\nu}}(s,a,b)\left[P_h(V_{h+1}^*-\underline{V}_{h+1}^{\mathrm{ref}})^2\right](s,a,b)}{n}}+\frac{c^2C^*SABH^3\iota}{n}$$

$$(V_{h+1}^*\geq\underline{V}_{h+1}\geq\underline{V}_{h+1}^{\mathrm{ref}})$$

$$=c^2\sqrt{SABH\iota}\sqrt{\frac{H^2C^*\sum_{h=1}^{H}\sum_{s}d_{h+1}^{\mu^*,\underline{\nu}}(s)(V_{h+1}^*(s)-\underline{V}_{h+1}^{\mathrm{ref}}(s))}{n}}+\frac{c^2C^*SABH^3\iota}{n}$$

$$\leq c^2\sqrt{SABH\iota}\sqrt{\frac{H^2C^*64\sqrt{\frac{C^*SABH^5\iota^2}{n_{\mathrm{ref}}}}}{n}}+\frac{c^2SABH^3C^*\iota}{n}\qquad\text{(Theorem B.5)}$$

$$=c^2\sqrt{\frac{192C^*SABH^3\iota\sqrt{C^*SABH^5\iota^2}}{n^{3/2}}}+\frac{c^2C^*SABH^3\iota}{n}$$

$$\leq\widetilde{O}\left(\sqrt{\frac{C^*SABH^3}{n}}\right).\qquad\qquad\qquad(n\geq C^*SABH^4)$$

$$\square$$

**Theorem C.9.** *Suppose Assumption 2.2 holds. For any $0<\delta<1$ and $n\geq C^*SABH^4$, with probability $1-\delta$, the output policy $\pi=(\underline{\mu},\overline{\nu})$ of Algorithm 1 satisfies*

$$V_1^*(s_1)-V_1^{\underline{\mu},*}(s_1)\leq\widetilde{O}\left(\sqrt{\frac{C^*SABH^3}{n}}\right),$$

$$V_1^{*,\overline{\nu}}(s_1)-V_1^*(s_1)\leq\widetilde{O}\left(\sqrt{\frac{C^*SABH^3}{n}}\right).$$

*As a result, we have*

$$\mathrm{Gap}(\underline{\mu},\overline{\nu})\leq\widetilde{O}\left(\sqrt{\frac{C^*SABH^3}{n}}\right).$$

*Proof.*

$$V_1^{\mu^*,\nu}(s_1) - \underline{V}_1(s_1)$$

$$\leq 2\mathbb{E}_{\mu^*,\nu}\sum_{h=1}^{H}\underline{b}_{h,0}(s_h,a_h,b_h) + 2\mathbb{E}_{\mu^*,\nu}\sum_{h=1}^{H}\underline{b}_{h,1}(s_h,a_h,b_h) \qquad \text{(Lemma C.4)}$$

$$\leq \widetilde{O}\left(\sqrt{\frac{C^*SABH^3}{n}}\sqrt{V_1^{\mu^*,\nu}(s_1) - \underline{V}_1(s_1)}\right) + \widetilde{O}\left(\sqrt{\frac{C^*SABH^3}{n}}\right)$$
$$\text{(Lemma C.7 and Lemma C.8)}$$

$$\leq \widetilde{O}\left(\sqrt{\frac{C^*SABH^3}{n}}\right) + \widetilde{O}\left(\frac{C^*SABH^3}{n}\right) \qquad \text{(Lemma E.5)}$$

$$= \widetilde{O}\left(\sqrt{\frac{C^*SABH^3}{n}}\right).$$

By the definition of NE, we have

$$V_1^*(s_1) - V_1^{\underline{\mu},*}(s_1) \leq V_1^{\mu^*,\nu}(s_1) - \underline{V}_1(s_1) \leq \widetilde{O}\left(\sqrt{\frac{C^*SABH^3}{n}}\right).$$

The second argument can be proven in a similar manner. Combining these two argument and we can prove that

$$\text{Gap}(\underline{\mu},\overline{\nu}) \leq \widetilde{O}\left(\sqrt{\frac{C^*SABH^3}{n}}\right).$$

$\square$

# D  Proofs in Section 4.3

## D.1  Uniform Coverage

**Theorem D.1.** *Suppose $d_m = \min\{d_h^\rho(s,a,b) : h \in [H], (s,a,b) \in \mathcal{S} \times \mathcal{A} \times \mathcal{B}\}$ and Assumption 2.2 holds. For any $0 < \delta < 1$, with probability $1 - \delta$, the output policy $\pi = (\underline{\mu},\overline{\nu})$ of Algorithm 1 satisfies*

$$V_1^*(s_1) - V_1^{\underline{\mu},*}(s_1) \leq 64\sqrt{\frac{H^5\iota^2}{nd_m}}, V_1^{*,\overline{\nu}}(s_1) - V_1^*(s_1) \leq 64\sqrt{\frac{H^5\iota^2}{nd_m}}.$$

*As a result, we have*

$$\text{Gap}(\underline{\mu},\overline{\nu}) \leq \widetilde{O}\left(\sqrt{\frac{H^5}{nd_m}}\right).$$

*Proof.* By Lemma B.3, with probability $1 - \delta$ we have

$$V_1^{\mu^*,*}(s_1) - V_1^{\underline{\mu},*}(s_1)$$

$$\leq 2\sum_{h=1}^{H}\mathbb{E}_{\mu^*,\nu}\underline{b}_h(s_h,a_h,b_h)$$

$$= 2\sum_{h=1}^{H}\mathbb{E}_{\mu^*,\nu}\left[4\sqrt{\frac{H^2\iota}{n_h(s,a,b)\vee 1}}\right]$$

$$\leq 2\sum_{h=1}^{H}\mathbb{E}_{\mu^*,\nu}\left[32\sqrt{\frac{H^3\iota^2}{nd_h^\rho(s,a,b)}}\right] \qquad \text{(Lemma B.1)}$$

$$=2\sum_{h=1}^{H}\sum_{(s,a,b)}d_h^{\mu^*,\underline{\nu}}(s,a,b)\left[32\sqrt{\frac{H^3\iota^2}{nd_h^\rho(s,a,b)}}\right]$$

$$\leq 64\sum_{h=1}^{H}\sum_{(s,a,b)}d_h^{\mu^*,\underline{\nu}}(s,a,b)\left[\sqrt{\frac{H^3\iota^2}{nd_m}}\right]$$

$$\leq 64\sqrt{\sum_{h=1}^{H}\sum_{(s,a,b)}d_h^{\mu^*,\underline{\nu}}(s,a,b)}\cdot\sqrt{\frac{\sum_{h=1}^{H}\sum_{(s,a,b)}d_h^{\mu^*,\underline{\nu}}(s,a,b)C^*H^3\iota^2}{nd_m}}$$

(Cauchy-Schwarz Inequality)

$$=\sqrt{H}\cdot\sqrt{\frac{H^4\iota^2}{nd_m}}$$

$$=64\sqrt{\frac{H^5\iota^2}{nd_m}}.$$

$\square$

**Theorem D.2.** *Suppose* $d_m=\min\{d_h^\rho(s,a,b):h\in[H],(s,a,b)\in\mathcal{S}\times\mathcal{A}\times\mathcal{B}\}$ *and Assumption 2.2 holds. For any* $0<\delta<1$ *and strategy* $\mu,\nu$, *with probability* $1-\delta$, *the pessimistic value* $\underline{V}_h$ *and optimistic estimate* $\overline{V}_h$ *of Algorithm 1 satisfies*

$$\mathbb{E}_{\mu^*,\nu}\left[V_h^*(s_h)-\underline{V}_h(s_h)\right]\leq 64\sqrt{\frac{H^5\iota^2}{nd_m}},\mathbb{E}_{\mu,\nu^*}\left[\overline{V}_h(s_h)-V_h^*(s_h)\right]\leq 64\sqrt{\frac{H^5\iota^2}{nd_m}},$$

*where* $s_h$ *is sampled from the trajectory following the strategy in the expectation at timestep* $h$.

*Proof.* By Lemma B.3, under good event $\mathcal{G}$ for all state $s$ we have

$$V_h^*(s)-V_h^{\underline{\mu},*}(s)\leq 2\sum_{t=h}^{H}\mathbb{E}_{\mu^*,\underline{\nu}}\left[\underline{b}_h(s_t,a_t,b_t)|s_h=s\right]$$

We define $\nu'=(\nu_1,\cdots,\nu_{h-1},\underline{\nu}_h,\cdots,\underline{\nu}_H)$. Then we have

$$\mathbb{E}_{\mu^*,\nu}\left[V_h^*(s_h)-\underline{V}_h(s_h)\right]\leq\mathbb{E}_{\mu^*,\nu}\left[2\sum_{t=h}^{H}\mathbb{E}_{\mu^*,\underline{\nu}}\left[\underline{b}_h(s_t,a_t,b_t)|s_h=s\right]|s\right]$$

$$=2\sum_{t=h}^{H}\mathbb{E}_{\mu^*,\nu'}\left[\underline{b}_h(s_t,a_t,b_t)\right].$$

Then following the proof of Theorem D.1, we can prove the argument.

$\square$

**Lemma D.3.** *Suppose* $d_m=\min\{d_h^\rho(s,a,b):h\in[H],(s,a,b)\in\mathcal{S}\times\mathcal{A}\times\mathcal{B}\}$ *and Assumption 3.1 holds. For* $n\geq H^3/d_m$, *we have*

$$\mathbb{E}_{\mu^*,\underline{\nu}}\sum_{h=1}^{H}\underline{b}_{h,0}(s_h,a_h,b_h)\leq\widetilde{O}\left(\sqrt{\frac{H^3}{nd_m}}\sqrt{V_1^{\mu^*,\underline{\nu}}(s_1)-\underline{V}_1(s_1)}\right)+\widetilde{O}\left(\sqrt{\frac{H^3}{nd_m}}\right).$$

*Proof.*

$$\mathbb{E}_{\mu^*,\underline{\nu}}\sum_{h=1}^{H}\underline{b}_{h,0}(s_h,a_h,b_h)$$

$$=c\mathbb{E}_{\mu^*,\underline{\nu}}\sum_{h=1}^{H}\left(\sqrt{\frac{\mathrm{Var}_{\widehat{P}_{h,0}(s,a,b)}(\underline{V}_{h+1}^{\mathrm{ref}})\iota}{n_{h,0}(s,a,b)\vee 1}}+\frac{H\iota}{n_{h,0}(s,a,b)\vee 1}\right)$$

$$\leq c\mathbb{E}_{\mu^*,\underline{\nu}}\sum_{h=1}^{H}\left(\sqrt{\frac{c\,\mathrm{Var}_{P_h(s,a,b)}(\underline{V}_{h+1}^{\mathrm{ref}})\iota}{nd_h^{\rho}(s,a,b)}}+\frac{cH\iota}{nd_h^{\rho}(s,a,b)}+\frac{cH\iota}{nd_h^{\rho}(s,a,b)}\right)$$

$$\leq c^2\sum_{h=1}^{H}\sum_{(s,a,b)}d_h^{\mu^*,\underline{\nu}}(s,a,b)\left(\sqrt{\frac{\mathrm{Var}_{P_h(s,a,b)}(\underline{V}_{h+1}^{\mathrm{ref}})\iota}{nd_m}}+\frac{H\iota}{nd_m}\right)$$

$$\leq c^2\sqrt{\sum_{h=1}^{H}\sum_{(s,a,b)}d_h^{\mu^*,\underline{\nu}}(s,a,b)}\left(\sqrt{\frac{\sum_{h=1}^{H}\sum_{(s,a,b)}d_h^{\mu^*,\underline{\nu}}(s,a,b)\,\mathrm{Var}_{P_h(s,a,b)}(\underline{V}_{h+1}^{\mathrm{ref}})\iota}{nd_m}}+\frac{H\iota}{nd_m}\right)$$

$$\text{(Cauchy-Schwarz inequality)}$$

$$\leq c^2\sqrt{H}\cdot\sqrt{\frac{\iota\sum_{h=1}^{H}\sum_{(s,a,b)}d_h^{\mu^*,\underline{\nu}}(s,a,b)\,\mathrm{Var}_{P_h(s,a,b)}(\underline{V}_{h+1}^{\mathrm{ref}})}{nd_m}}+\frac{c^2 H\iota}{nd_m}$$

$$\leq c^2\sqrt{H\iota}\cdot\sqrt{\frac{\sum_{h=1}^{H}\mathbb{E}_{\mu^*,\underline{\nu}}\left[\mathrm{Var}_{P_h(s,a,b)}(\underline{V}_{h+1}^{\mathrm{ref}})\right]}{nd_m}}+\frac{c^2 H\iota}{nd_m}$$

$$\leq c^2\sqrt{H\iota}\cdot\sqrt{\frac{\sum_{h=1}^{H}\mathbb{E}_{\mu^*,\underline{\nu}}\left[\mathrm{Var}_{P_h(s,a,b)}(V_{h+1}^{\mu^*,\underline{\nu}})+2H[P_h(V_{h+1}^{\mu^*,\underline{\nu}}-\underline{V}_{h+1}^{\mathrm{ref}})](s,a,b)\right]}{nd_m}}+\frac{c^2 H\iota}{nd_m}$$

$$\text{(Lemma E.4)}$$

$$\leq c^2\sqrt{H\iota}\cdot\sqrt{\frac{H^2+2H\sum_{h=1}^{H}\mathbb{E}_{\mu^*,\underline{\nu}}\left[V_{h+1}^{\mu^*,\underline{\nu}}(s_{h+1})-\underline{V}_{h+1}^{\mathrm{ref}}(s_{h+1})\right]}{nd_m}}+\frac{c^2 H\iota}{nd_m}\qquad\text{(Lemma C.5)}$$

$$= c^2\sqrt{H\iota}\cdot\sqrt{\frac{H^2+2H\sum_{h=1}^{H}\mathbb{E}_{\mu^*,\underline{\nu}}\left[V_{h+1}^{\mu^*,\underline{\nu}}(s_{h+1})-V_{h+1}^*(s_{h+1})+V_{h+1}^*(s_{h+1})-\underline{V}_{h+1}^{\mathrm{ref}}(s_{h+1})\right]}{nd_m}}+\frac{c^2 H\iota}{nd_m}$$

$$\leq c^2\sqrt{H\iota}\cdot\sqrt{\frac{H^2+2H^2(V_1^{\mu^*,\underline{\nu}}(s_1)-\underline{V}_1(s_1))+128H\sqrt{\frac{H^5\iota^2}{n_{\mathrm{ref}}d_m}}}{nd_m}}+\frac{c^2 H\iota}{nd_m}$$

$$\text{(Lemma C.6 and Theorem D.2)}$$

$$\leq \frac{c^2\sqrt{H^3\iota}}{\sqrt{nd_m}}+\frac{c^2\sqrt{384H^2\iota\sqrt{H^5\iota^2}}}{(nd_m)^{3/4}}+\frac{c\sqrt{2H^3\iota}}{\sqrt{nd_m}}\sqrt{V_1^{\mu^*,\underline{\nu}}(s_1)-\underline{V}_1(s_1)}+\frac{c^2 H\iota}{nd_m}$$

$$\leq \widetilde{O}\left(\sqrt{\frac{H^3}{nd_m}}\sqrt{V_1^{\mu^*,\underline{\nu}}(s_1)-\underline{V}_1(s_1)}\right)+\widetilde{O}\left(\sqrt{\frac{H^3}{nd_m}}\right).\qquad\qquad (n\geq H^3/d_m)$$

$$\square$$

**Lemma D.4.** *For $n\geq H^4/d_m$, we have*

$$\mathbb{E}_{\mu^*,\underline{\nu}}\sum_{h=1}^{H}\underline{b}_{h,1}(s_h,a_h,b_h)\leq\widetilde{O}\left(\sqrt{\frac{H^3}{nd_m}}\right).$$

*Proof.*

$$\mathbb{E}_{\mu^*,\underline{\nu}}\sum_{h=1}^{H}\underline{b}_{h,1}(s_h,a_h,b_h)$$

$$=c\mathbb{E}_{\mu^*,\underline{\nu}}\sum_{h=1}^{H}\left(\sqrt{\frac{\mathrm{Var}_{\widehat{P}_{h,0}(s,a,b)}(\underline{V}_{h+1}-\underline{V}_{h+1}^{\mathrm{ref}})\iota}{n_{h,1}(s,a,b)\vee 1}}+\frac{H\iota}{n_{h,1}(s,a,b)\vee 1}\right)$$

$$\leq c\mathbb{E}_{\mu^*,\underline{\nu}}\sum_{h=1}^{H}\left(\sqrt{\frac{cH\,\mathrm{Var}_{P_h(s,a,b)}(\underline{V}_{h+1}-\underline{V}_{h+1}^{\mathrm{ref}})\iota}{nd_h^\rho(s,a,b)}}+\frac{cH^2\iota}{nd_h^\rho(s,a,b)}+\frac{cH^2\iota}{nd_h^\rho(s,a,b)}\right)$$

$$\leq c^2\mathbb{E}_{\mu^*,\underline{\nu}}\sum_{h=1}^{H}\left(\sqrt{\frac{H\left[P_h(\underline{V}_{h+1}-\underline{V}_{h+1}^{\mathrm{ref}})^2\right](s,a,b)\iota}{nd_h^\rho(s,a,b)}}+\frac{H^2\iota}{nd_h^\rho(s,a,b)}\right)$$

$$\leq c^2\sum_{h=1}^{H}\sum_{(s,a,b)}d_h^{\mu^*,\underline{\nu}}(s,a,b)\left(\sqrt{\frac{H\left[P_h(\underline{V}_{h+1}-\underline{V}_{h+1}^{\mathrm{ref}})^2\right](s,a,b)\iota}{nd_m}}+\frac{H^2\iota}{nd_m}\right)$$

$$\leq c^2\sqrt{\sum_{h=1}^{H}\sum_{(s,a,b)}d_h^{\mu^*,\underline{\nu}}(s,a,b)}\left(\sqrt{\frac{\sum_{h=1}^{H}\sum_{(s,a,b)}Hd_h^{\mu^*,\underline{\nu}}(s,a,b)\left[P_h(\underline{V}_{h+1}-\underline{V}_{h+1}^{\mathrm{ref}})^2\right](s,a,b)\iota}{nd_m}}+\frac{H^2\iota}{nd_m}\right)$$

$$\text{(Cauchy-Schwarz Inequality)}$$

$$\leq c^2\sqrt{H}\sqrt{\frac{H\iota\sum_{h=1}^{H}\sum_{(s,a,b)}d_h^{\mu^*,\underline{\nu}}(s,a,b)\left[P_h(\underline{V}_{h+1}-\underline{V}_{h+1}^{\mathrm{ref}})^2\right](s,a,b)}{nd_m}}+\frac{c^2H^3\iota}{nd_m}$$

$$\leq c^2\sqrt{H\iota}\sqrt{\frac{H\iota\sum_{h=1}^{H}\sum_{(s,a,b)}d_h^{\mu^*,\underline{\nu}}(s,a,b)\left[P_h(V_{h+1}^*-\underline{V}_{h+1}^{\mathrm{ref}})^2\right](s,a,b)}{nd_m}}+\frac{c^2H^3\iota}{nd_m}$$

$$(V_{h+1}^*\geq \underline{V}_{h+1}\geq \underline{V}_{h+1}^{\mathrm{ref}})$$

$$=c^2\sqrt{H\iota}\sqrt{\frac{H^2\sum_{h=1}^{H}\sum_s d_{h+1}^{\mu^*,\underline{\nu}}(s)(V_{h+1}^*(s)-\underline{V}_{h+1}^{\mathrm{ref}}(s))}{nd_m}}+\frac{c^2H^3\iota}{nd_m}$$

$$\leq c^2\sqrt{H\iota}\sqrt{\frac{H^264\sqrt{\frac{H^5\iota^2}{n_{\mathrm{ref}}d_m}}}{nd_m}}+\frac{c^2H^3\iota}{nd_m}\qquad\text{(Theorem D.2)}$$

$$=c^2\sqrt{\frac{192H^3\iota\sqrt{H^5\iota^2}}{(nd_m)^{3/2}}}+\frac{c^2H^3\iota}{nd_m}$$

$$\leq \widetilde{O}\left(\sqrt{\frac{H^3}{nd_m}}\right).\qquad (n\geq H^4/d_m)$$

$\square$

**Theorem D.5.** *Suppose $d_m=\min\{d_h^\rho(s,a,b):h\in[H],(s,a,b)\in\mathcal{S}\times\mathcal{A}\times\mathcal{B}\}$ and Assumption 3.1 holds. For any $0<\delta<1$ and $n\geq H^4/d_m$, with probability $1-\delta$, the output policy $\pi=(\underline{\mu},\overline{\nu})$ of Algorithm 2 satisfies*

$$V_1^*(s_1)-V_1^{\underline{\mu},*}(s_1)\leq \widetilde{O}\left(\sqrt{\frac{H^3}{nd_m}}\right),\;V_1^{*,\overline{\nu}}(s_1)-V_1^*(s_1)\leq \widetilde{O}\left(\sqrt{\frac{H^3}{nd_m}}\right).$$

*As a result, we have*

$$\mathrm{Gap}(\underline{\mu},\overline{\nu})\leq \widetilde{O}\left(\sqrt{\frac{H^3}{nd_m}}\right).$$

*Proof.*

$$V_1^{\mu^*,\underline{\nu}}(s_1) - \underline{V}_1(s_1)$$

$$\leq 2\mathbb{E}_{\mu^*,\underline{\nu}} \sum_{h=1}^H \underline{b}_{h,0}(s_h, a_h, b_h) + 2\mathbb{E}_{\mu^*,\underline{\nu}} \sum_{h=1}^H \underline{b}_{h,1}(s_h, a_h, b_h) \qquad \text{(Lemma C.4)}$$

$$\leq \widetilde{O}\left(\sqrt{\frac{H^3}{nd_m}}\sqrt{V_1^{\mu^*,\underline{\nu}}(s_1) - \underline{V}_1(s_1)}\right) + \widetilde{O}\left(\sqrt{\frac{H^3}{nd_m}}\right) \qquad \text{(Lemma D.3 and Lemma D.4)}$$

$$\leq \widetilde{O}\left(\sqrt{\frac{H^3}{nd_m}}\right) + \widetilde{O}\left(\frac{H^3}{nd_m}\right) \qquad \text{(Lemma E.5)}$$

$$= \widetilde{O}\left(\sqrt{\frac{H^3}{nd_m}}\right).$$

By the definition of NE, we have

$$V_1^*(s_1) - V_1^{\underline{\mu},*}(s_1) \leq V_1^{\mu^*,\underline{\nu}}(s_1) - \underline{V}_1(s_1) \leq \widetilde{O}\left(\sqrt{\frac{H^3}{nd_m}}\right).$$

The second argument can be proven in a similar manner. Combining two arguments together and we can derive that

$$\text{Gap}(\underline{\mu}, \overline{\nu}) \leq \widetilde{O}\left(\sqrt{\frac{H^3}{nd_m}}\right).$$

□

## D.2 Turn-based Markov Games

For turn-based Markov games, there always exists a pure (deterministic) NE equilibrium strategy. As a result, we can have that $\mu^*, \nu^*, \underline{\mu}, \underline{\nu}, \overline{\mu}, \overline{\nu}$ are all pure strategy.

**Theorem D.6.** *Suppose Assumption 2.2 holds. For any $0 < \delta < 1$, with probability $1 - \delta$, the output policy $\pi = (\underline{\mu}, \overline{\nu})$ of Algorithm 1 satisfies*

$$V_1^*(s_1) - V_1^{\underline{\mu},*}(s_1) \leq 64\sqrt{\frac{C^* S H^5 \iota^2}{n}}, V_1^{*,\overline{\nu}}(s_1) - V_1^*(s_1) \leq 64\sqrt{\frac{C^* S H^5 \iota^2}{n}}.$$

*As a result, we have*

$$\text{Gap}(\underline{\mu}, \overline{\nu}) \leq \widetilde{O}\left(\sqrt{\frac{C^* S H^5}{n}}\right).$$

*Proof.* By Lemma B.3, with probability $1 - \delta$ we have

$$V_1^{\mu^*,*}(s_1) - V_1^{\underline{\mu},*}(s_1)$$

$$\leq 2\sum_{h=1}^H \mathbb{E}_{\mu^*,\underline{\nu}} \underline{b}_h(s_h, a_h, b_h)$$

$$= 2\sum_{h=1}^H \mathbb{E}_{\mu^*,\underline{\nu}} \left[4\sqrt{\frac{H^2\iota}{n_h(s,a,b) \vee 1}}\right]$$

$$\leq 2\sum_{h=1}^H \mathbb{E}_{\mu^*,\underline{\nu}} \left[32\sqrt{\frac{H^3\iota^2}{nd_h^\rho(s,a,b)}}\right] \qquad \text{(Lemma B.1)}$$

$$=2\sum_{h=1}^{H}\sum_{(s,a,b)}d_h^{\mu^*,\underline{\nu}}(s,a,b)\left[32\sqrt{\frac{H^3\iota^2}{nd_h^\rho(s,a,b)}}\right]$$

$$\leq64\sum_{h=1}^{H}\sum_{(s,a,b)}\left[\sqrt{\frac{d_h^{\mu^*,\underline{\nu}}(s,a,b)C^*H^3\iota^2}{n}}\right]$$

$$=64\sum_{h=1}^{H}\sum_{s\in\mathcal{S}}\left[\sqrt{\frac{d_h^{\mu^*,\underline{\nu}}(s,\mu^*(s),\underline{\nu}(s))C^*H^3\iota^2}{n}}\right]\qquad(\mu^*,\underline{\nu}\text{ are deterministic strategy.})$$

$$\leq64\sqrt{SH}\cdot\sqrt{\frac{\sum_{h=1}^{H}\sum_{s\in\mathcal{S}}d_h^{\mu^*,\underline{\nu}}(s,\mu^*(s),\underline{\nu}(s))C^*H^3\iota^2}{n}}\qquad(\text{Cauchy-Schwarz Inequality})$$

$$=64\sqrt{\frac{C^*SH^5\iota^2}{n}}.$$

$\square$

**Theorem D.7.** *Suppose Assumption 2.2 holds. For any $0<\delta<1$ and policy $\mu,\nu$, with probability $1-\delta$, the pessimistic value $\underline{V}_h$ of Algorithm 1 satisfies*

$$\mathbb{E}_{\mu^*,\nu}\left[V_h^*(s_h)-\underline{V}_h(s_h)\right]\leq64\sqrt{\frac{C^*SH^5\iota^2}{n}},$$

$$\mathbb{E}_{\mu,\nu^*}\left[\overline{V}_h(s_h)-V_h^*(s_h)\right]\leq64\sqrt{\frac{C^*SH^5\iota^2}{n}},$$

*where $s_h$ is sampled from the trajectory following the strategy in the expectation at timestep $h$.*

*Proof.* By Lemma B.3, under good event $\mathcal{G}$ for all state $s$ we have

$$V_h^*(s)-V_h^{\mu,*}(s)$$

$$\leq2\sum_{t=h}^{H}\mathbb{E}_{\mu^*,\underline{\nu}}\left[\underline{b}_h(s_t,a_t,b_t)|s_h=s\right]$$

We define $\nu'=(\nu_1,\cdots,\nu_{h-1},\underline{\nu}_h,\cdots,\underline{\nu}_H)$. Then we have

$$\mathbb{E}_{\mu^*,\nu}\left[V_h^*(s_h)-\underline{V}_h(s_h)\right]\leq\mathbb{E}_{\mu^*,\nu}\left[2\sum_{t=h}^{H}\mathbb{E}_{\mu^*,\underline{\nu}}\left[\underline{b}_h(s_t,a_t,b_t)|s_h=s\right]|s\right]$$

$$=2\sum_{t=h}^{H}\mathbb{E}_{\mu^*,\nu'}\left[\underline{b}_h(s_t,a_t,b_t)\right].$$

Then following the proof of Theorem D.6, we can prove the argument.

$\square$

**Lemma D.8.** *For $n\geq C^*SH^3$, we have*

$$\mathbb{E}_{\mu^*,\underline{\nu}}\sum_{h=1}^{H}\underline{b}_{h,0}(s_h,a_h,b_h)\leq\widetilde{O}\left(\sqrt{\frac{C^*SH^3}{n}}\sqrt{V_1^{\mu^*,\underline{\nu}}(s_1)-\underline{V}_1(s_1)}\right)+\widetilde{O}\left(\sqrt{\frac{C^*SH^3}{n}}\right).$$

*Proof.*

$$\mathbb{E}_{\mu^*,\underline{\nu}}\sum_{h=1}^{H}\underline{b}_{h,0}(s_h,a_h,b_h)$$

$$=c\mathbb{E}_{\mu^*,\underline{\nu}}\sum_{h=1}^{H}\left(\sqrt{\frac{\text{Var}_{\widehat{P}_{h,0}(s,a,b)}(V_{h+1}^{\text{ref}})\iota}{n_{h,0}(s,a,b)\vee1}}+\frac{H\iota}{n_{h,0}(s,a,b)\vee1}\right)$$

$$
\begin{aligned}
\leq & c\mathbb{E}_{\mu^*,\underline{\nu}} \sum_{h=1}^{H} \left( \sqrt{\frac{c\,\mathrm{Var}_{P_h(s,a,b)}(\underline{V}_{h+1}^{\mathrm{ref}})\iota}{nd_h^{\rho}(s,a,b)}} + \frac{cH\iota}{nd_h^{\rho}(s,a,b)} + \frac{cH\iota}{nd_h^{\rho}(s,a,b)} \right) \\
= & c^2 \sum_{h=1}^{H} \sum_{s\in\mathcal{S}} d_h^{\mu^*,\underline{\nu}}(s,\mu^*(s),\underline{\nu}(s)) \left( \sqrt{\frac{\mathrm{Var}_{P_h(s,\mu^*(s),\underline{\nu}(s))}(\underline{V}_{h+1}^{\mathrm{ref}})\iota}{nd_h^{\rho}(s,\mu^*(s),\underline{\nu}(s))}} + \frac{H\iota}{nd_h^{\rho}(s,\mu^*(s),\underline{\nu}(s))} \right) \\
\leq & c^2 \sum_{h=1}^{H} \sum_{s\in\mathcal{S}} \left( \sqrt{\frac{C^* d_h^{\mu^*,\underline{\nu}}(s,\mu^*(s),\underline{\nu}(s))\,\mathrm{Var}_{P_h(s,\mu^*(s),\underline{\nu}(s))}(\underline{V}_{h+1}^{\mathrm{ref}})\iota}{n}} + \frac{C^* H\iota}{n} \right)
\end{aligned}
$$

$(\mu^*, \underline{\nu}$ are deterministic strategies.$)$

$$
\begin{aligned}
\leq & c^2 \sqrt{SH} \cdot \sqrt{\frac{C^*\iota \sum_{h=1}^{H}\sum_{s\in\mathcal{S}} d_h^{\mu^*,\underline{\nu}}(s,\mu^*(s),\underline{\nu}(s))\,\mathrm{Var}_{P_h(s,\mu^*(s),\underline{\nu}(s))}(\underline{V}_{h+1}^{\mathrm{ref}})}{n}} + \frac{c^2 SC^* H\iota}{n} \\
\leq & c^2 \sqrt{C^* SH\iota} \cdot \sqrt{\frac{\sum_{h=1}^{H} \mathbb{E}_{\mu^*,\underline{\nu}}\left[\mathrm{Var}_{P_h(s,a,b)}(\underline{V}_{h+1}^{\mathrm{ref}})\right]}{n}} + \frac{c^2 SC^* H\iota}{n} \\
\leq & c^2 \sqrt{C^* SH\iota} \cdot \sqrt{\frac{\sum_{h=1}^{H} \mathbb{E}_{\mu^*,\underline{\nu}}\left[\mathrm{Var}_{P_h(s,a,b)}(V_{h+1}^{\mu^*,\underline{\nu}}) + 2H[P_h(V_{h+1}^{\mu^*,\underline{\nu}} - \underline{V}_{h+1}^{\mathrm{ref}})](s,a,b)\right]}{n}} + \frac{c^2 SC^* H\iota}{n}
\end{aligned}
$$

(Lemma E.4)

$$
\leq c^2 \sqrt{C^* SH\iota} \cdot \sqrt{\frac{H^2 + 2H\sum_{h=1}^{H}\mathbb{E}_{\mu^*,\underline{\nu}}\left[V_{h+1}^{\mu^*,\underline{\nu}}(s_{h+1}) - \underline{V}_{h+1}^{\mathrm{ref}}(s_{h+1})\right]}{n}} + \frac{c^2 SC^* H\iota}{n}
$$

(Lemma C.5)

$$
= c^2 \sqrt{C^* SH\iota} \cdot \sqrt{\frac{H^2 + 2H\sum_{h=1}^{H}\mathbb{E}_{\mu^*,\underline{\nu}}\left[V_{h+1}^{\mu^*,\underline{\nu}}(s_{h+1}) - V_{h+1}^*(s_{h+1}) + V_{h+1}^*(s_{h+1}) - \underline{V}_{h+1}^{\mathrm{ref}}(s_{h+1})\right]}{n}} + \frac{c^2 SC^* H\iota}{n}
$$

$$
\leq c^2 \sqrt{C^* SH\iota} \cdot \sqrt{\frac{H^2 + 2H^2(V_1^{\mu^*,\underline{\nu}}(s_1) - \underline{V}_1(s_1)) + 128H\sqrt{\frac{C^* SH^5\iota^2}{n_{\mathrm{ref}}}}}{n}} + \frac{c^2 SC^* H\iota}{n}
$$

(Lemma C.6 and Theorem D.7)

$$
\begin{aligned}
\leq & \frac{c^2 \sqrt{C^* SH^3\iota}}{\sqrt{n}} + \frac{c^2 \sqrt{384 C^* SH^2\iota\sqrt{C^* SH^5\iota^2}}}{n^{3/4}} + \frac{c\sqrt{2C^* SH^3\iota}}{\sqrt{n}}\sqrt{V_1^{\mu^*,\underline{\nu}}(s_1) - \underline{V}_1(s_1)} + \frac{c^2 SC^* H\iota}{n} \\
\leq & \widetilde{O}\left( \sqrt{\frac{C^* SH^3}{n}}\sqrt{V_1^{\mu^*,\underline{\nu}}(s_1) - \underline{V}_1(s_1)} \right) + \widetilde{O}\left( \sqrt{\frac{C^* SH^3}{n}} \right). \qquad (n \geq C^* SH^3)
\end{aligned}
$$

$\square$

**Lemma D.9.** *For $n \geq C^* SH^4$, we have*

$$
\mathbb{E}_{\mu^*,\underline{\nu}} \sum_{h=1}^{H} \underline{b}_{h,1}(s_h,a_h,b_h) \leq \widetilde{O}\left( \sqrt{\frac{C^* SH^3}{n}} \right).
$$

*Proof.*

$$
\begin{aligned}
& \mathbb{E}_{\mu^*,\underline{\nu}} \sum_{h=1}^{H} \underline{b}_{h,1}(s_h,a_h,b_h) \\
= & c\mathbb{E}_{\mu^*,\underline{\nu}} \sum_{h=1}^{H} \left( \sqrt{\frac{\mathrm{Var}_{\widehat{P}_{h,0}(s,a,b)}(\underline{V}_{h+1} - \underline{V}_{h+1}^{\mathrm{ref}})\iota}{n_{h,1}(s,a,b) \vee 1}} + \frac{H\iota}{n_{h,1}(s,a,b) \vee 1} \right)
\end{aligned}
$$

$$\leq c\mathbb{E}_{\mu^*,\underline{\nu}} \sum_{h=1}^{H} \left( \sqrt{\frac{cH \operatorname{Var}_{P_h(s,a,b)}(\underline{V}_{h+1} - \underline{V}_{h+1}^{\mathrm{ref}})\iota}{nd_h^\rho(s,a,b)}} + \frac{cH^2\iota}{nd_h^\rho(s,a,b)} + \frac{cH^2\iota}{nd_h^\rho(s,a,b)} \right)$$

$$\leq c^2\mathbb{E}_{\mu^*,\underline{\nu}} \sum_{h=1}^{H} \left( \sqrt{\frac{H\left[P_h(\underline{V}_{h+1} - \underline{V}_{h+1}^{\mathrm{ref}})^2\right](s,a,b)\iota}{nd_h^\rho(s,a,b)}} + \frac{H^2\iota}{nd_h^\rho(s,a,b)} \right)$$

$$= c^2 \sum_{h=1}^{H} \sum_{s\in\mathcal{S}} d_h^{\mu^*,\underline{\nu}}(s,\mu^*(s),\underline{\nu}(s)) \left( \sqrt{\frac{H\left[P_h(\underline{V}_{h+1} - \underline{V}_{h+1}^{\mathrm{ref}})^2\right](s,\mu^*(s),\underline{\nu}(s))\iota}{nd_h^\rho(s,\mu^*(s),\underline{\nu}(s))}} + \frac{H^2\iota}{nd_h^\rho(s,\mu^*(s),\underline{\nu}(s))} \right)$$

$$\leq c^2 \sum_{h=1}^{H} \sum_{s\in\mathcal{S}} \left( \sqrt{\frac{C^*Hd_h^{\mu^*,\underline{\nu}}(s,\mu^*(s),\underline{\nu}(s))\left[P_h(\underline{V}_{h+1} - \underline{V}_{h+1}^{\mathrm{ref}})^2\right](s,\mu^*(s),\underline{\nu}(s))\iota}{n_1}} + \frac{H^2C^*\iota}{n_1} \right)$$

(Cauchy-Schwarz Inequality)

$$\leq c^2\sqrt{SH\iota}\sqrt{\frac{C^*H\sum_{h=1}^{H}\sum_{s\in\mathcal{S}} d_h^{\mu^*,\underline{\nu}}(s,\mu^*(s),\underline{\nu}(s))\left[P_h(\underline{V}_{h+1} - \underline{V}_{h+1}^{\mathrm{ref}})^2\right](s,\mu^*(s),\underline{\nu}(s))}{n}} + \frac{c^2SH^3C^*\iota}{n}$$

$$\leq c^2\sqrt{SH\iota}\sqrt{\frac{C^*H\iota\sum_{h=1}^{H}\sum_{(s,a,b)} d_h^{\mu^*,\underline{\nu}}(s,a,b)\left[P_h(V_{h+1}^* - \underline{V}_{h+1}^{\mathrm{ref}})^2\right](s,a,b)}{n}} + \frac{c^2C^*SH^3\iota}{n}$$

($V_{h+1}^* \geq \underline{V}_{h+1} \geq \underline{V}_{h+1}^{\mathrm{ref}}$)

$$= c^2\sqrt{SH\iota}\sqrt{\frac{H^2C^*\sum_{h=1}^{H}\sum_s d_{h+1}^{\mu^*,\underline{\nu}}(s)(V_{h+1}^*(s) - \underline{V}_{h+1}^{\mathrm{ref}}(s))}{n}} + \frac{c^2C^*SH^3\iota}{n}$$

$$\leq c^2\sqrt{SH\iota}\sqrt{\frac{H^2C^*64\sqrt{\frac{C^*SH^5\iota^2}{n_{\mathrm{ref}}}}}{n}} + \frac{c^2SH^3C^*\iota}{n}$$

(Theorem D.7)

$$= c^2\sqrt{\frac{192C^*SH^3\iota\sqrt{C^*SH^5\iota^2}}{n^{3/2}}} + \frac{c^2C^*SH^3\iota}{n}$$

$$\leq \widetilde{O}\left( \sqrt{\frac{C^*SH^3}{n}} \right).$$

($n \geq C^*SH^4$)

□

**Theorem D.10.** *Suppose Assumption 2.2 holds for a turn-based Markov game and $n \geq C^*SH^4$. For any $0 < \delta < 1$, with probability $1 - \delta$, the output policy $\pi = (\underline{\mu}, \overline{\nu})$ of Algorithm 1 satisfies*

$$V_1^*(s_1) - V_1^{\underline{\mu},*}(s_1) \leq \widetilde{O}\left( \sqrt{\frac{C^*SH^3}{n}} \right),$$

$$V_1^{*,\overline{\nu}}(s_1) - V_1^*(s_1) \leq \widetilde{O}\left( \sqrt{\frac{C^*SH^3}{n}} \right).$$

*As a result, we have*

$$\mathrm{Gap}(\underline{\mu},\overline{\nu}) \leq \widetilde{O}\left( \sqrt{\frac{C^*SH^3}{n}} \right).$$

*Proof.*

$$V_1^{\mu^*,\underline{\nu}}(s_1) - \underline{V}_1(s_1)$$

$$\leq 2\mathbb{E}_{\mu^*,\underline{\nu}} \sum_{h=1}^{H} \underline{b}_{h,0}(s_h, a_h, b_h) + 2\mathbb{E}_{\mu^*,\underline{\nu}} \sum_{h=1}^{H} \underline{b}_{h,1}(s_h, a_h, b_h) \qquad \text{(Lemma C.4)}$$

$$\leq \widetilde{O}\left(\sqrt{\frac{C^*SH^3}{n}} \sqrt{V_1^{\mu^*,\underline{\nu}}(s_1) - \underline{V}_1(s_1)}\right) + \widetilde{O}\left(\sqrt{\frac{C^*SH^3}{n}}\right) \quad \text{(Lemma D.8 and Lemma D.9)}$$

$$\leq \widetilde{O}\left(\sqrt{\frac{C^*SH^3}{n}}\right) + \widetilde{O}\left(\frac{C^*SH^3}{n}\right) \qquad \text{(Lemma E.5)}$$

$$= \widetilde{O}\left(\sqrt{\frac{C^*SH^3}{n}}\right).$$

By the definition of NE, we have

$$V_1^*(s_1) - V_1^{\underline{\mu},*}(s_1) \leq V_1^{\mu^*,\underline{\nu}}(s_1) - \underline{V}_1(s_1) \leq \widetilde{O}\left(\sqrt{\frac{C^*SH^3}{n}}\right).$$

The second argument can be proven in a similar manner. Combining these two arguments and we can derive that

$$\text{Gap}(\underline{\mu}, \overline{\nu}) \leq \widetilde{O}\left(\sqrt{\frac{C^*SH^3}{n}}\right).$$

$\square$

# E  Auxiliary Lemmas

**Lemma E.1.** *(Multiplicative Chernoff bound). Let $X$ be a binomial random variable with parameter $p$, $n$. For any $1 \geq \theta > 0$, we have that*

$$\mathbb{P}[(1-\theta)pn < X < (1+\theta)pn] < 2e^{-\frac{\theta^2 pn}{2}}$$

**Lemma E.2.** *For all $(s_h, a_h, b_h) \in \mathcal{K}_h$ and any $\|V\|_\infty \leq H$, with probability $1 - \delta$ we have*

$$\sqrt{\underset{\widehat{P}^\dagger_{s_h,a_h,b_h}}{\text{Var}}(V)} \leq \sqrt{\underset{P^\dagger_{s_h,a_h,b_h}}{\text{Var}}(V)} + cH\sqrt{\frac{\iota}{nd_h^\mu(s_h, a_h, b_h)}}.$$

*Proof.* The is a direct application of Lemma E.3 with a union bound. $\square$

**Lemma E.3.** *(Empirical Berstein Inequality [Maurer and Pontil, 2009]) Let $n \geq 2$ and $V \in \mathbb{R}^S$ be any functions with $\|V\|_\infty \leq H$, $P$ be any $S$-dimensional distribution and $\widehat{P}$ be its empirical version using $n$ samples. Then with probability $1 - \delta$,*

$$\left|\sqrt{\underset{\widehat{P}}{\text{Var}}(V)} - \sqrt{\frac{n-1}{n}\underset{P}{\text{Var}}(V)}\right| \leq 2H\sqrt{\frac{\log(2/\delta)}{n-1}}.$$

**Lemma E.4.** *For $0 \leq V \leq V' \leq H$, we have*

$$\underset{P_h(s,a,b)}{\text{Var}}(V) \leq \underset{P_h(s,a,b)}{\text{Var}}(V') + 2H[P_h(V' - V)](s,a,b).$$

*Proof.*

$$\underset{P_h(s,a,b)}{\text{Var}}(V) - \underset{P_h(s,a,b)}{\text{Var}}(V')$$
$$\leq \left[P_h(V)^2 - (P_hV)^2 - P_h(V')^2 + (P_hV')^2\right](s,a,b)$$
$$= \left[P_h(V+V')(V-V') + [P_h(V'-V)][P_h(v'+v)]\right](s,a,b)$$
$$\leq 2H[P_h(V'-V)](s,a,b).$$

$\square$

**Lemma E.5.** *If $x \le a\sqrt{x} + b$ for $a, b > 0$, then we have*

$$x \le 2a^2 + 2b.$$

*Proof.* We have

$$\left(\sqrt{x} - \frac{a}{2}\right)^2 \le b + \frac{a^2}{4}.$$

If $\sqrt{x} < \frac{a}{2}$, the argument holds directly. Otherwise we have

$$\sqrt{x} - \frac{a}{2} \le \sqrt{b + \frac{a^2}{4}} \le \sqrt{b} + \frac{a}{2}.$$

So we have $\sqrt{x} \le \sqrt{b} + a$, which implies $x \le 2(a^2 + b)$. $\qquad\square$