# OpenReview forum: "When are Offline Two-Player Zero-Sum Markov Games Solvable?"
_NeurIPS.cc/2022/Conference — NeurIPS 2022 Accept_

### Official Review · Reviewer_6SL9 · 2022-06-29

**Rating:** 7
**Confidence:** 3
**Soundness:** 3 good
**Presentation:** 3 good
**Contribution:** 3 good

**Summary:**

The paper studies the relevant offline MARL setting, which is relevant in practice, where only a small amount of data are given to the RL algorithm to infer a policy. The main contribution of the paper is to provide the necessary and sufficient assumption for learning a NE strategy in the offline Markov Game setting. This assumption is different from the one used in the single agent. In fact, in MDPs, the necessary assumption implies that one optimal policy is "covered" in the given dataset. However, in MGs, we need to see also the interaction between the NE policy and all the other agent's policies.
After this result, the authors construct an algorithm based on pessimism that is provably efficient under this assumption.

**Questions:**

See Weaknesses.

**Limitations:**

The paper is theoretical, so the main limitations are due to the applicability of the proposed method in practice.

1. The state and action spaces are finite.
2. The algorithm needs to compute at each step a NE for each state and time-step.
3. The paper does not provide an experimental evaluation of the proposed algorithm.

I think all of these limitations are reasonable since the main contribution of the paper is theoretical.

**Strengths And Weaknesses:**

Strengths

1. The paper, as far as I know, is the first one that tracts the offline zero-sum Markov Game problem.

2. In the paper a necessary and sufficient condition for learning in offline zero-sum MG problems is given. The idea behind the proof is quite simple but, although the techniques used are not novel, the result is important for the ML community.

3. The authors provide an algorithm that is provably efficient (although it does not match the lower bound in general).

4. The paper is in general well-written and easy to follow. The authors spent space giving intuition about the different proof of the different theorems. I find this very helpful for a reader who does not want to read the full appendix.

Weaknesses

I think the paper has no important weaknesses. However, I list below the weaknesses.

1. In the paper the authors prove that the Unilateral concentration assumption is the sufficient and necessary assumption to learn in offline MGs problem. However, the paper will become stronger if the authors would give some results that bound the approximation of the NE that the proposed algorithm can achieve when this assumption does not hold and when only the Single strategy concentration assumption holds. Do the authors have an intuition about it?

2. Section 4.1 gives a first result when Hoeffding confidence intervals are used. However, the authors show that with Bernstein's confidence intervals they can improve the previous result. So, why do we need section 4.1? I suggest removing this section and putting in the main paper only Algorithm 2. Moreover, it is important that the reader does not have to go to the appendix to read the main algorithm.

3. From sections 4.1 and 4.2, it is not clear what are the main challenges the authors faced to translate an offline RL algorithm to MGs. Could the authors list these technical challenges in their answer?

---

> ### Author Response · Authors · 2022-07-31
> **Response to Reviewer 6SL9**
>
> Thanks for your appreciation! We address your questions below.
>
> - **Weaker Assumption:** This is a very interesting question! Generally, it is difficult to find even an approximate-NE with single strategy concentration (as a direct corollary from Theorem 3.3). However, it is possible for other structured games like potential games and identical-interest games to enjoy a weaker assumption. In addition, if an (approximate-)NE (approximately) satisfies the unilateral concentration assumption, we believe an approximate-NE can be recovered and we leave it to future work.
> - **Removing Section 4.1:** You are correct that the results in Section 4.2 are stronger than those in Section 4.1. We write Section 4.1 to illustrate the algorithm framework as it is simple and more intuitive. We will downweigh or remove it in the final version. Thanks for your suggestion!
> - **Technical Challenges:** The main technical contributions are listed in Section 1.1 (line 73 to line 78). We develop the monotonic update and self-bounding technique, which are described around line 270 and line 274. These two techniques are not used in previous single-agent offline RL papers and are new to offline zero-sum Markov games due to the interaction between two players.

---

> > ### Comment · Reviewer_6SL9 · 2022-08-03
> > **Answer to authors**
> >
> > Thanks to the authors for their answers to my questions.
> >
> > After reading the rebuttal and other reviews I continue to think that the paper is a relevant contribution to the conference.
> >
> > I only encourage the authors to remove section 4.1 and move it to the appendix.

---

### Official Review · Reviewer_tuL7 · 2022-07-01

**Rating:** 6
**Confidence:** 2
**Soundness:** 3 good
**Presentation:** 2 fair
**Contribution:** 3 good

**Summary:**

The paper proposes the unilateral concentration condition which provides a necessary and sufficient dataset coverage to learn an NE using offline methods. The paper proposes an algorithm that converges to an NE under these conditions and is minimax optimal.

**Questions:**

Q1: Would "full concentration" be a better term to use? "Uniform" seems to imply some sort of uniformity over policy or state space, but really it means full-support. Although, I suppose this term has been inherited from the single-agent RL community?


**Limitations:**

The main limitations "uniform concentration" or "sequential Makov" were clearly stated. Scaling factors is also discussed.

**Strengths And Weaknesses:**

Offline learning in multiagent domains is an important worthwhile area of study. Two-player zero-sum is an acceptable area to start.

I found the first half of the paper well written. The second was difficult to follow (although my ignorance was a contributing a factor).

Soundness:

I am strongly convinced that the "unilateral concentration condition" claims are true. They seem intuitively true and the proofs are convincing.

I am also convinced of the impossibility results.

The claims around the optimal offline algorithm are plausible. While I did not spot any errors in the proofs, the techniques used in this section are beyond my area of expertise and unfortunately the time constraints on reviewing did not allow me to fully grasp this section as thoroughly as I would have liked. I hope the other reviewers can vouch for the soundness of this section.

Comments:

It may be worth noting that the uniform concentration condition, while trivial, is not that uncommon in practice. For example, data generated using an epsilon-greedy strategy, common in value based methods, or policy gradient methods which are necessarily stochastic would generate data that is uniform concentration.

Rating:

I will be willing to improve my rating if I felt more confident on the Section 4's soundness. I will remain open to other reviewers opinions on this matter and will be ready to improve my score with consensus. [Edit: consensus is that Section 4 is sound, improving my score from 4->6]

Minor comment:
* Line 25: “...two-players sequentially select actions…” - In Markov games players take actions simultaneously (over multiple time steps).

---

> ### Author Response · Authors · 2022-07-31
> **Response to Reviewer tuL7**
>
> Thanks for your careful reading! We address your concerns below.
>
> - **Soundness of Section 4:** We believe Section 4 has no technical issue and other reviewers have no question on that as well. We would be more than grateful if you could improve your score if you believe there is no soundness issue.
> - **Full Concentration/Uniform Concentration:** You are right about this. The word `Uniform Concentration` is inherited from single-agent RL community and we use it just for consistency.
> - **Uniform concentration is common:** we agree that uniform concentration condition is available in many scenarios. On the other hand, our paper aims to find the weakest assumption. We will be clear about this point in the final version.

---

### Official Review · Reviewer_wGFm · 2022-07-11

**Rating:** 7
**Confidence:** 4
**Soundness:** 4 excellent
**Presentation:** 4 excellent
**Contribution:** 3 good

**Summary:**

This paper studies the learning of two player zero-sum Markov games from data. The results introduce a new condition "Unilateral concentration" which informally requires data of plays when one of the player follows a NE equilibrium. The authors show that this is necessary and sufficient condition for learnability. The authors provide a provable bound poly time algorithm under the condition, one weaker bound using Hoeffding inequality and another using empirical Bernstein bounds.


**Questions:**

N/A

**Strengths And Weaknesses:**

The writing is clear and the results look solid to me. The proof sketches are helpful. I really did not find anything to complain other than few minor points that I state for a more rounded result. I must admit I am not very well versed in offline RL literature but the paper does a good job at comparing with related work.

It would have been good to have a few (even small) experiments - as the work is about learnability from data, it is surprising to not have any experiments.

Little better notation would make things easier to understand, especially for the variance of value function.

The second contribution in intro says "necessary and sufficient dataset coverage assumption for solving offline Markov games" it is important to be state zero-sum here - I know the paper title has zero-sum but this claim is broader.

---

> ### Author Response · Authors · 2022-07-31
> **Response to Reviewer wGFm**
>
> Thanks for your appreciation and suggestions! We have changed the wording in our paper.

---

> ### Author Response · Authors · 2022-08-07
> **Any other questions to be addressed？**
>
> Thanks again for your review. We hope our answers could increase your confidence. As the discussion period is close to the end and we have not yet heard back from you, we would be glad to see if our rebuttal response has addressed your concerns questions/concerns.
> We are more than happy to discuss further if you have any further concerns and issues, please kindly let us know your feedback. Thank you for your time and help!

---

### Official Review · Reviewer_LCjP · 2022-07-11

**Rating:** 8
**Confidence:** 2
**Soundness:** 4 excellent
**Presentation:** 4 excellent
**Contribution:** 3 good

**Summary:**

The paper proves a new assumption named unilateral concentration assumption as a necessary and sufficient dataset coverage condition for achieving optimal (NE) strategies in offline Markov game setting. A new Nash VI-based algorithm with pessimism is also proposed to achieve the minimax lower bound when the game is turn-based or under a stronger uniform concentration assumption. The results are novel and interesting with significant theoretical value.

**Questions:**

Is the sentence "A direct corollary is that uniform concentration (Assumption 2.1) is not sufficient for NE learning." in Sec. 3 a typo? Assumption 2.1 is single strategy concentration.

In page 8 l260, the 'ref' subscripts are missed for reference function.

For data splitting scheme, is it the same as random sample without replacement for each timestep with a fixed sample buffer size? How will sampling with replacement affect the final results?

**Limitations:**

As is also noted by the authors, the remaining factor $AB$ in the bound of PNVI is unclear to be removable or not.


**Strengths And Weaknesses:**

The paper has a clear description, arguments and proofs about the proposed assumptions and theorems. The motivation is clear. A sharper Bernstein bound is applied with a reference advantage technique from previous paper to make PNVI match the lower bound.

For the argument that no weaker assumption than unilateral concentration can ensure any algorithm to learn NE is true under the hypothesis that this is claimed for all games in the game class. And the proof is conducted by constructing a hard instance. However, from a more practical view, an assumption for a given game would allow a weaker assumption, since most games in the game class are not the hard instance. To this end, it would be interesting to see some weaker assumptions allowing existing any algorithm to learn NE for a certain game that is not hard instance. After all, the data coverage in practice is very hard to guarantee.

---

> ### Author Response · Authors · 2022-07-31
> **Response to Reviewer LCjP**
>
> Thanks for your appreciation! We address your questions below.
> - **Weaker Assumption:** This is a very interesting question! Generally, it is difficult to find even an approximate-NE with single strategy concentration (as a direct corollary from Theorem 3.3). However, it is possible for other structured games like potential games and identical-interest games to enjoy a weaker assumption. In addition, if an (approximate-)NE (approximately) satisfies the unilateral concentration assumption, we believe an approximate-NE can still be recovered and we leave it to future work.
> - **Typo (uniform concentration):** Thanks for pointing out the typo. This should be single strategy concentration.
> - **Data Splitting:** We use random sampling without replacement to ensure that $\widehat{P}_h$ and $\overline{V}_{h+1}$ are independent. Using sampling with replacement may induce dependence between different timesteps.

---

> > ### Comment · Reviewer_LCjP · 2022-08-07
> > **Response to Author Rebuttal**
> >
> > Thanks for the reply. I have no further question at this point.

---

> ### Author Response · Authors · 2022-08-07
> **Any other questions to be addressed？**
>
> Thanks again for your review. We hope our answers could increase your confidence. As the discussion period is close to the end and we have not yet heard back from you, we would be glad to see if our rebuttal response has addressed your concerns questions/concerns.
> We are more than happy to discuss further if you have any further concerns and issues, please kindly let us know your feedback. Thank you for your time and help!

---

### Author Response · Authors · 2022-07-31
**General comments**

We have changed all the minor wording issues in the main paper. Please feel free to check out.

---

### Meta-Review · Area_Chair_WkP8 · 2022-08-26

**Recommendation:** Accept
**Confidence:** Certain

**Metareview:**

This paper is a clear accept, with very positive reviews overall. I trust that the reviewers will address the minor concerns raised by the reviewers in their final version.

**Award:**

No

---

### Decision · Program_Chairs · 2022-09-14

Accept